# Craig-Gordon model validation using stable isotope ratios in water vapor over the Southern Ocean

Shaakir Shabir Dar[1], Prosenjit Ghosh[1,2], Ankit Swaraj[2], and Anil Kumar[3]

[1]Centre for Earth Sciences, Indian Institute of Science, Bengaluru, 560012, Karnataka, India.
[2]Divecha Centre for Climate Change, Indian Institute of Science, Bengaluru, 560012, Karnataka, India.
[3]National Centre for Polar and Ocean Research, Vaso-da-Gama, 403804, Goa, India.

**Correspondence:** Prosenjit Ghosh (pghosh@iisc.ac.in)

**Abstract.** The stable isotopic composition of water vapor over a water body is governed by the isotopic composition of surface water, ambient vapor isotopic composition, exchange and mixing processes at the water-air interface as well as the local meteorological conditions. These parameters form inputs to the Craig-Gordon models, used for predicting the isotopic composition of vapor produced from the surface water due to the evaporation process. In this study we present water vapor, surface water isotope ratios and meteorological parameters across latitudinal transects in the Southern Ocean ($27.38°S$ to $69.34°S$ and $21.98°S$ to $66.8°$) during two austral summers. The performance of Traditional Craig-Gordon (TCG) (Craig and Gordon, 1965) and the Unified Craig-Gordon (UCG) (Gonfiantini et al., 2018) models is evaluated to predict the isotopic composition of evaporated water vapor flux in the diverse oceanic settings. The models are run for the molecular diffusivity ratios suggested by (Merlivat, 1978) (MJ), (Cappa et al., 2003) (CD) and (Pfahl and Wernli, 2009) (PW) and different turbulent indices (x) i.e. fractional contribution of molecular vs turbulent diffusion. It is found that the $UCG^{MJ}_{x=0.8}, UCG^{CD}_{x=0.6}, TCG^{MJ}_{x=0.6}$ and $TCG^{CD}_{x=0.7}$ models predicted the isotopic composition that best matches with the observations. The relative contribution from locally generated and advected moisture is calculated at the water vapor sampling points, along the latitudinal transects, assigning the representative end member isotopic compositions and by solving the two-component mixing model. The results suggest varying contribution of advected westerly component with an increasing trend up to 65°S. Beyond 65°S, the proportion of Antarctic moisture was found to be prominent and increasing linearly towards the coast.

## 1   Introduction

The knowledge of factors governing the evaporation of water from the oceans is an essential part of our understanding of the hydrological cycle. The oceans regulate the climate of the earth through heat and moisture transport (Chahine, 1992). Nearly $\approx 97\%$ of the water of earth is in the oceans as saline while the residual $\approx 3\%$ is fresh water stored in groundwater, glaciers and lakes, or flowing as rivers and streams (Korzoun and Sokolov, 1978). Evaporation of ocean water generates vapour and forms the initial reservoir for circulation in the hydrological cycle. A fraction of this vapor, only $\approx 10\%$ of it is transported inland to generate precipitation, while rest of the moisture precipitates over the ocean during its transit (Oki and Kanae, 2006; Shiklomanov, 1998).

Measurements of the isotope composition of water in the various reservoirs of the hydrological cycle operating over the oceans is useful to infer information about the origin of water masses and understanding the formation mechanisms, transport pathways and finally the precipitation processes (Craig, 1961; Dansgaard, 1964; Yoshimura, 2015; Gat, 1996; Araguás-Araguás et al., 2000; Noone and Sturm, 2010; Gat et al., 2003; Benetti et al., 2014; Galewsky et al., 2016). Comparatively large volume of data exists over land to understand the terrestrial hydrological cycle, through the Global Network in Precipitation (GNIP) initiative of the International Atomic Energy Agency (IAEA). However, only a handful records on the spatial and temporal variability of precipitation and vapor isotopic composition over the oceans is available for any assessment (e.g. Gat et al. (2003); Uemura et al. (2008); Benetti et al. (2015, 2017b, a); Rahul et al. (2018); Prasanna et al. (2018); Bonne et al. (2019)). Hence, further effort is needed to enhance the spatial and temporal sampling coverage over the oceans.

The isotopic composition of vapor on top of a water body is governed by the factors: i) Thermodynamic equilibrium process for phase transformation at a particular temperature ii) Kinetic or non-equilibrium processes where role of relative humidity and wind is significant and iii) Large-scale transport and mixing: due to the movement of air parcels laterally and vertically. Craig and Gordon (1965) initially proposed a two-layer model to simulate the isotopic composition of evaporated (referred to as the Traditional Craig-Gordon model). Recently, Gonfiantini et al. (2018) put forward a modified version referred to as the Unified Craig-Gordon Model. Both of these models incorporate the equilibrium and kinetic processes to simulate the isotopic composition of evaporated moisture. However, in order to get a realistic picture of the hydrological cycle over the ocean, the horizontal transport/advective mixing is important and should be incorporated.

In this paper we present stable isotope ratios in water vapor and ocean surface water from different locations covering varied oceanic settings; i.e. tropical, subtropical and polar latitudes, with a large range in the sea surface temperature, relative humidity and wind speed. While the role of temperature dependent equilibrium fractionation is well understood, the role of kinetic processes is under debate and requires further scrutiny. The performance of these Craig-Gordon evaporation models to simulate the isotopic composition of evaporation flux is evaluated along the sampling transect for different molecular diffusivity ratios and different fractions of molecular vs turbulent diffusion in the framework of the global closure assumption. The evaporation flux by the Craig-Gordon models is calculated assuming the 'global closure' i.e. the isotopic composition of atmospheric vapor is equal to the isotopic composition of evaporation. The models and the conditions that best match with the observations are identified, which are then used to calculate the local evaporation flux. This as done in the context of estimating the contribution of advected vs in-situ derived vapor along the sampling transect assuming a complete mixing of the advected and the locally generated vapor in the sampled water vapor in our study.

## 2 Methods

### 2.1 Sampling, isotopic analysis and meteorological parameters

The samples (water vapor, and surface water) for this study were collected along the stretch from Mauritius to Prydz Bay (24°S to 69°S and 57°E to 76°E) during two successive austral summers (January 2017 (SOE-IX) and December 2017 to January 2018 (SOE-X)) onboard the ocean research vessel SA Agulhas. The water vapor sampling inlet was set at ≈ 15m above the sea

level. An aggregate of 71 water vapor samples were collected during the two expeditions. Fig. 1 shows the water vapor sampling locations. Alongside water vapor, 49 surface water samples were also collected. The details about the sampling procedures for collection of water vapor and surface water samples are given in the supplementary document. All these subjected to isotopic analysis using Finnigan Gasbench peripheral connected with an isotope ratio mass spectrometer (ThermoScientific MAT 253) (details are provided in the supplementary document). The isotope ratios are expressed in ‰ using the standard $\delta$ notation relative to Vienna Standard Mean Ocean Water (VSMOW).

In addition to water sampling, relative humidity (h), wind speed (ws), air temperature (Ta), sea surface temperature (SST), and atmospheric pressure (P) was recorded continuously during the expedition. Fig. 2 shows the latitudinal variation of these meteorological parameters. A wide range of these physical conditions are encountered since the sampling encompasses a large latitudinal transect.

## 2.2 Backward air-mass trajectories

In order to reconstruct the vertical profile of the atmospheric moisture transport along the sampling transect, backward air mass trajectories were generated using the Hybrid Single Particle Lagrangian Integrated Trajectory (HYSPLIT) model (Draxler and Hess, 1998; Stein et al., 2015) of NOAA-NCEP/NCAR forced with the Reanalysis data (Kanamitsu et al., 2002). HYSPLIT is a computational model hybrid between Lagrangian and Eulerian methods which generates the paths traversed by the air parcels and calculates meteorological variables such as temperature, relative humidity, specific humidity, rainfall, pressure etc. along the route. Back trajectories for 3 days are extracted since the average residence time of atmospheric moisture over the oceans is ≈3 days Trenberth (1998); Van Der Ent and Tuinenburg (2017). Figure 3 shows the back trajectories for the water vapor sampling locations. The sampling locations can be broadly categorized into zones which are defined by different wind patterns (i.e. velocity and the moisture carrying capacity). Westerlies and polar easterlies were identified based these 72 hour back-trajectories constructed at three different heights above the ocean surface. During the SOE X expedition, the change in trajectories to westerlies was at ≈31°S. At ≈63°S, change from westerlies to polar easterlies is seen. For SOE IX the transition from the westerlies to easterlies and then to polar westerlies was documented at the ≈33°S and ≈64°S latitudes respectively.

## 2.3 The Craig-Gordon Models

Craig-Gordon in 1965 (CG) Craig and Gordon (1965) proposed the first theoretical model to explain the isotopic composition of water vapor during the evaporation process. The isotopic composition of vapor generated on top of the ocean water depends on the isotopic composition of the surface oceanic water, the isotopic composition of water vapor in the ambient atmosphere along with the relative humidity at the site of sample collection. The interplay of equilibrium and kinetic fractionation between these phases governs the final isotopic composition in the water vapour and liquid. The equilibrium fractionation between ocean water and vapor is controlled by the sea surface temperature (SST). In comparison, relative humidity and wind speed control the the kinetic fractionation through the combination of processes which include both molecular and turbulent diffusion. Molecular diffusion leads to isotopic fractionation between liquid and vapor whereas the turbulent diffusion is non-fractionating. To estimate the isotopic composition of water vapor CG model invokes two-layers; a laminar layer above the air-water interface

where the transport process is active via molecular diffusion and a turbulent layer above the laminar layer in which the molecular transfer is predominantly by the action of turbulent diffusion. Assuming there is no divergence/convergence of air mass over the oceanic atmosphere, the isotopic ratio of the evaporation flux is given as Craig and Gordon (1965) referred to as Traditional Craig-Gordon Model (TCG):

$$R_{ev} = \alpha_k . \frac{R_L . \alpha_{eq} - h . R_A}{1 - h} \tag{1}$$

Where $R_L$, $R_A$, $h$, $\alpha_k$ and $\alpha_{eq}$ are respectively, the isotopic composition of the liquid water, the isotopic composition environmental atmospheric moisture, relative humidity, the kinetic and the equilibrium fractionation factors. The TCG models in this form and with modifications have been employed in diverse applications and used in numerous studies. The 'global closure' i.e. assuming a steady state is achieved in which the isotopic composition of vapor removed from the system has the same composition as atmospheric vapor (Merlivat, 1978):

$$R_A = R_{ev} \tag{2}$$

the global closure assumption (Eq. 2), is substituted in Eq (1) to give;

$$R_{ev} = \alpha_k . \frac{R_L . \alpha_{eq} - h . R_{ev}}{1 - h} \tag{3}$$

$$R_{ev}(1 - h) = \alpha_k . [R_L . \alpha_{eq} - h . R_{ev}] \tag{4}$$

$$R_{ev}(1 - h) + \alpha_k h . R_{ev} = \alpha_k . R_L . \alpha_{eq} \tag{5}$$

$$R_{ev}[(1 - h) + \alpha_k . h] = \alpha_k . R_L . \alpha_{eq} \tag{6}$$

$$R_{ev} = \frac{\alpha_{eq} \alpha_k R_L}{(1 - h) + \alpha_k . h} \tag{7}$$

Recently, Gonfiantini et al. (2018) proposed a modified version of the model, termed as Unified Craig-Gordon (UCG) model in which the parameters controlling the isotopic composition of the evaporation flux are considered simultaneously. From Gonfiantini et al. (2018), the net evaporation rate of liquid water (E) is the difference between the vaporization rate, $\psi_{vap}$, and the atmospheric vapor capture rate (i.e; condensation) by the liquid water, $\psi_{cap}$.

$$E = \psi_{vap} - \psi_{cap} = (\gamma - h)\psi_{cap}^o \tag{8}$$

Where the $\psi^o_{cap}$ is the vaporization rate of pure water, h is the relative humidity and $\gamma$ is the thermodynamic activity coefficient of evaporating water which is <1 for the saline solutions and 1 for the pure water or dilute solutions.

From Eq. (8), We can write;

$$R_{ev}(\gamma - h)\Psi^o_{vap} = R_{esc}\gamma\Psi^o_{vap} - R_{cap}h\Psi^o_{vap} \tag{9}$$

$$R_{ev}(\gamma - h)\Psi^o_{vap} = \frac{R_L}{\alpha_{eq}\alpha^x_{diff}}\gamma\Psi^o_{vap} - \frac{R_A}{\alpha^x_{diff}}h\Psi^o_{vap} \tag{10}$$

$$R_{ev} = \frac{\frac{R_L}{\alpha_{eq}\alpha^x_{diff}}\gamma - \frac{R_A}{\alpha^x_{diff}}h}{\gamma - h} \tag{11}$$

Where $R_L$, $R_{esc}$, $R_{cap}$ and $R_A$ are, respectively the isotopic composition of the liquid water, isotopic composition of vapor
escaping to the saturated layer above which is in thermodynamic equilibrium with water, isotopic composition of environmental atmospheric moisture captured by the equilibrium layer and the isotopic composition environmental atmospheric moisture. $R_L$, $R_{esc}$, $R_{cap}$ and $R_A$ are defined as in Gonfiantini et al. (2018). $\alpha_{eq}$ is the isotopic fractionation factor between the liquid water and the vapor in the equilibrium layer. $\alpha_{diff}$ is the isotopic fractionation factor for diffusion in air affecting the vapor escaping from the equilibrium layer and the environmental vapor entering the equilibrium layer; $x$ is the turbulent index of atmosphere.
Introducing the global closure assumption, Eq. (2) in Eq. (11) gives:

$$R_{ev} = \frac{\frac{R_L}{\alpha_{eq}\alpha^x_{diff}}\gamma - \frac{R_{ev}}{\alpha^x_{diff}}h}{\gamma - h} \tag{12}$$

$$R_{ev}(\gamma - h) = \frac{R_L}{\alpha_{eq}\alpha^x_{diff}}\gamma - \frac{R_{ev}}{\alpha^x_{diff}}h \tag{13}$$

$$R_{ev}(\gamma - h) + \frac{R_{ev}}{\alpha^x_{diff}}h = \frac{R_L}{\alpha_{eq}\alpha^x_{diff}}\gamma \tag{14}$$

$$R_{ev}[(\gamma - h) + \frac{h}{\alpha^x_{diff}}] = \frac{R_L}{\alpha_{eq}\alpha^x_{diff}}\gamma \tag{15}$$

$$R_{ev}[\frac{(\gamma - h)\alpha^x_{diff} + h}{\alpha^x_{diff}}] = \frac{R_L}{\alpha_{eq}\alpha^x_{diff}}\gamma \tag{16}$$

$$R_{ev} = \frac{R_L}{\alpha_{eq}\alpha^x_{diff}}\gamma[\frac{\alpha^x_{diff}}{(\gamma - h)\alpha^x_{diff} + h}] \tag{17}$$

$$R_{ev} = \frac{R_L \gamma}{\alpha_{eq}[\alpha_{diff}^x(\gamma - h) + h]} \tag{18}$$

The temperature dependent equilibrium fractionation factor is calculated using the formulation given by (Horita and Wesolowski, 1994). The kinetic factor takes into account diffusion in air affecting the vapor escaping from the equilibrium layer and is controlled by $\alpha_{diff}$, which is the molecular diffusivity of the different isotopologues of water. Molecular diffusivities ($\alpha_{diff}^O$, $\alpha_{diff}^H$) data are taken from three previous studies Merlivat (1978) (1.0285, 1.0251), Cappa et al. (2003) (1.0318, 1.0164) and Pfahl and Wernli (2009) (1.0076, 1.0039) referred to as MJ, CD and PW respectively. x is the turbulence index of atmosphere which signifies the proportion of vapor that escapes by isotopic fractionating molecular diffusion and non-fractionating turbulent diffusion. When x = 1 the vapor escapes solely by molecular diffusion and for x = 0 the vapor escapes only due to turbulent diffusion.

## 3 Results

### 3.1 Isotopic measurements along the transect

$\delta^{18}O$ of surface water was > 0 ‰ until $\approx 40°$S latitude. A transition to lighter isotopic composition was observed beyond $\approx$ 45°S latitude with a drop documented in the surface water isotopic values on approaching the coastal Antarctic regions. Figure 4a shows the latitudinal variation of $\delta^{18}O_{sw}$, plotted along with salinity values measured along the transect. In addition, the $\delta^{18}O$ of ocean surface water extracted from the Global Sea Water 18O Database (SWD) (Schmidt et al., 1999) are also plotted. There is a mismatch between the observed depleted isotopic values near coastal Antarctica with SWD values. The SWD is a surface interpolated dataset based on point observations in the global ocean. This is probably one of the major causes of the difference, the others being the season or the month of sample collection.

The $\delta^{18}O_{wv}$ and $\delta^2H_{wv}$ in water vapor samples showed a consistent trend across latitude for both the expeditions. The $\delta^{18}O_{wv}$ ($\delta^2H_{wv}$) of water vapor varies from -10.9‰ (-80.8‰) to -27.5‰(-221.4‰) respectively. The vapor isotopic composition is seen to be gradually decreasing with lighter isotopic values at higher latitudes. A steady drop was noted from $\approx 30°$S to $\approx 65°$S and a sharp change in the gradient was registered at $\approx 65°$S. Extreme lighter values recorded on approaching $\approx 65°$S are attributed to factors such as low temperature and the mixing of lighter vapor from continental Antarctica (Uemura et al., 2008). There are deviations from this general trend with heavier isotopic composition observed at the higher latitudes or vice versa. These variations can be accounted, by taking into consideration the source and the path of air masses. The lighter (heavier) values of vapor isotopic composition can be traced to the source being lower (higher) latitudes.

Deuterium excess (d-excess or dxs), defined as $d - excess = \delta^2H - 8 \times \delta^{18}O$, is a second order isotope parameter which is a measure of kinetic fractionation during evaporation (Dansgaard, 1964). d-excess in the water vapor correlates with meteorological parameters at the ocean surface such as relative humidity, sea surface temperature and wind speed (Uemura et al., 2008; Rahul et al., 2018; Benetti et al., 2014; Midhun et al., 2013). Therefore, it serves as a proxy for the moisture source conditions in the evaporation regions. The dxs and relative humidity are strongly coupled, which is determined by the magnitude of

moisture gradient between evaporating water surface and overlying unsaturated air. In other words, lower the relative humidity higher is the dxs in the overlying moisture. Wind speed regulates the turbulent vs molecular diffusion across the diffusive layer.

The role of SST in governing the dxs is through the process of equilibrium fractionation, which is temperature dependent. The dxs values in water vapor samples range from 18.7‰ to -23.7‰. A relatively higher dxs values in the water vapor from ≈25°S to ≈45°S with a slight step change to lower dxs values was recorded on approaching 45°S which extends until ≈65°S. Beyond ≈ 65°S a slight increment in the vapor dxs was observed. The very low dxs values due to mixing of vapor evaporated from sea-spray under high wind speed conditions are observed during the passage of extra-tropical cyclones. The statistics of the

isotopic composition of water vapor are tabulated in Table 1.

## 3.2   Meteorological controls on the isotopic composition of water vapor

The $\delta^{18}O_{wv}$ and $\delta^2H_{wv}$ are positively correlated with SST, negatively correlated with wind speed and uncorrelated with relative humidity. For all the water vapor samples, $\delta^2H_{wv}$ and $\delta^{18}O_{wv}$ are correlated with SST explaining ≈33% of the variance in $\delta^{18}O_{wv}$ and ≈50% of the variance in $\delta^2H_{wv}$. The correlation coefficient is higher if sampling from individual

175   years is considered separately. In all cases, the slope and intercept of the regression equation between the isotopic composition of water vapor and SST is comparable with previous observations from the Southern Ocean Uemura et al. (2008). The linear regression plots are shown Figure 5 and the regression parameters (slope, intercept, standard errors and $r^2$) for $\delta^{18}O$ and $\delta^2H$ are listed in Table 1.(S) and Table 2.(S) respectively. The regression equations are calculated for different sample classifications, with and without the influence of Antarctic vapor mixing as evident from the back trajectories (i.e. samples collected north of

65°S) and for individual expeditions.

Figure 6 shows the regression plots of dxs in vapor with the meteorological conditions and the parameters defining the regression equations are listed in Table 2. For samples collected north of 65°S, the linear regression equation describing the relationship between dxs and relative humidity is dxs=-0.56h+46.36 ($r^2$=0.49). These slope and intercept values are similar to the earlier records, documenting the isotope variability in water vapour from the Southern Ocean (Uemura et al., 2008; Rahul

et al., 2018) the Bay of Bengal (Midhun et al., 2013), the Atlantic (Benetti et al., 2014) and the Mediterranean (Gat et al., 2003). For samples collected south of 65°S the relationship becomes weaker. The strength of the dxs vs h relationship was stronger if data exclusively from the expeditions is considered separately, for the SOE IX and SOE X as dxs=-0.64h+57.4 ($r^2$=0.77) and dxs=-0.64h+48.7 ($r^2$=0.61) respectively. Collectively for both the expeditions, the dxs in vapor is positively correlated with the SST and the regression parameters are comparable with those from previous observations in the Southern Ocean and

also for the Atlantic Ocean and the Bay of Bengal. For SST vs dxs, the linear regression equation for samples collected north of 65°S is given by dxs=0.70sst-4.65 ($r^2$=0.49). The dxs of water vapor samples are negatively correlated with wind speed. For samples collected north of 65°S the correlation the regression equation is given by dxs=-0.53ws+11.65 ($r^2$=0.23). Our observation is consistent with the earlier studies suggesting the dependency of water vapor d-excess on relative humidity, SST and wind speed.

# 4  Discussion

## 4.1  Craig-Gordon (CG) model evaluation

The isotopic composition of evaporation flux from the oceans is calculated using the CG models (TCG and UCG) assuming three molecular diffusivity ratios driving the kinetic fractionation and for varied contribution of turbulent vs molecular diffusion enabled transport factors. The simulated values of the isotopic composition of evaporation flux with these different models under the global closure assumption are compared with the measured isotopic values of water vapor over the ocean. The model and the constraints that best describe the observations are selected based on the model predicted and observed relationships between the dxs of water vapor and physical parameters (SST, ws and h).

The TCG and the UCG models are run for MJ, CD and PW molecular diffusivities and for the turbulence index of the atmosphere varying from 0-1 with an increment of 0.1. Figure 7 and Figure 8 shows the comparison between the TCG and UCG modelled vapor isotopic composition ($\delta^{18}O$ and d-excess) with the observations. There are values for the turbulence index (x) of the atmosphere where model predicted $\delta^{18}O$ and d-excess overlap with the observations for both TCG and UCG models with MJ and CD molecular diffusivity ratios. However, there is a clear mismatch between the model predicted $\delta^{18}O$ and d-excess for the recommended PW molecular diffusivities in both UCG and TCG models. Another noteworthy feature of the plots is the for all the model runs a large difference is seen between the modelled and observed isotopic composition for water vapor samples collected south of $\approx 65°$S latitudes and the best match is seen for samples collected North of $\approx 65°$S. This difference is attributed to the advection and mixing of lighter Antarctic moisture to local moisture for samples collected beyond $\approx 65°$S.

To evaluate the performance of the prediction by these models and identify the parameters that best describe the observations, the slope of the dxs vs relative humidity predicted by the different model runs are compared with the observed relationships documented based on actual data on samples collected north of 65°S. Figure 1.(S) depicts the comparison between the observed and the model predicted relationships. The UCG models and the parameters that match the observed slope of the relative humidity vs d-excess relationship ($-0.56 \pm 0.08$) are $UCG^{MJ}_{x=0.8}$, $UCG^{CD}_{x=0.6}$ and $UCG^{PW}_{x=0}$. Similarly for the TCG models $TCG^{MJ}_{x=0.6}$, $TCG^{CD}_{x=0.7}$ and $TCG^{PW}_{x=0}$ predict the slopes that are comparable with the observed value. The $\delta^{18}O$ and d-excess of predicted by these models are plotted with the observations in Figure 9.

The consistency of model results and observations are best described using a linear regression equation which links model predicted d-excess and the meteorological parameters (relative humidity, sea surface temperature and wind speed). These regression plots are displayed in Figure 2.(S). The difference between the model predicted and the observed values of slopes and intercepts are shown in Figure 10. The largest difference between the observed and model predicted slopes are intercepts are for the PW molecular diffusivities for both UCG and the TCG models and therefore excluded from further discussion. For the dxs vs relative humidity relationship, $UCG^{MJ}_{x=0.8}$ and $UCG^{CD}_{x=0.6}$ show the smallest difference between the observed and modelled slopes and intercepts followed by $TCG^{MJ}_{x=0.6}$ and $TCG^{CD}_{x=0.7}$. In case of dxs vs SST relationship, 'the TCG models show the least difference between the slopes and the UCG model predicts the intercept values that are consistent with the observations. Similarly, for the and dxs vs ws relationships, the UCG and the TCG models produce the values that predict the

slope and the intercept values with the least deviation from the observed values. The models that best describe the slope and intercept values of linear regression equation defining the d-excess vs the meteorological parameters, the root mean square error of the modelled vs observed $\delta^{18}O$ and d-excess are listed in Table 3. The ability of the models to predict the $\delta^{18}O$ and d-excess are better demonstrated by the water vapor samples which were collected north of 65°S. The models predict the d-excess with a better correlation than $\delta^{18}O$ and the TCG model show a slightly higher possibility to predict the d-excess values than the UCG model.

## 4.2   Understanding the equilibrium/disequilibrium

The isotopic composition of water vapor over the ocean is governed by the equilibrium and kinetic processes which are defined by the meteorological condition. However, considering only these factors is insufficient to explain the observed variation in the isotopic composition of vapor on top of the ocean. Advective mixing of transported vapor to the locally generated vapor is important and needs to be taken into consideration. Fig. 11a shows the difference between the $\delta^{18}O$ and $\delta^2 H$ isotopic composition of vapor (at equilibrium with ocean surface water) and the observed vapor isotopic composition. Kinetic fractionation can explain a part of the departure from the equilibrium state and is evaluated based on the Craig-Gordon models as described in the previous section ($E_{UCG}^{MJ,0.8}$, $E_{UCG}^{CD,0.6}$, $E_{TCG}^{MJ,0.6}$ and $E_{TCG}^{CD,0.7}$). The difference between isotopic composition of equilibrium vapor ($\delta^{18}O$ and $\delta^2 H$) and the modelled isotopic composition by the $E_{UCG}$, $E_{TCG}$ is also plotted in Figure 11b-e. In order to calculate the fractional contribution of the local and advected moisture along the sampling transect, a two component mixing framework is invoked. The local end member is based on the isotopic composition of vapor predicted by the best match UCG and the TCG model predicted parameters. The calculations are done assuming the isotopic composition of the advected vapor due to westerlies similar to the earlier proposition (Uemura et al., 2008) ($\delta^2 H \approx$-109‰) in the region between 31°S to 65°S. For samples collected in the polar ocean south of 65°S, the temperature plays the role of limiting the local evaporation process and hence the large differences from the equilibrium conditions can be explained by invoking the process of mixing of Antarctic vapor which is transported to this region by the interplay of polar easterlies. The average isotopic composition of water vapor collected at Dome C site (Dec 2014-Jan) (Wei et al., 2019) ($\delta^2 H$= -490±23‰) is chosen as representative of the advected vapor transported by the polar easterlies. It is seen that in order to explain the water vapor isotope ratio observation over the ocean south of 65°S, the contribution of lighter Antarctic vapor is expected. Fig. 12b shows the relative contribution of advected and locally generated moisture in our observation. The advected component is a prominent component of the ambient vapor on approaching higher latitudes. South of 65°S the amount of moisture present in the atmosphere is less and is largely local in origin with a small mixing of lighter Antarctic vapor. However, the contribution of the Antarctic vapor linearly increases on approaching the coastal regions.

## 5   Conclusions

In this study, the isotopic composition of water vapor and surface water samples collected across a latitudinal transect from Mauritius to Prydz in the Southern Ocean are described. The isotopic composition of evaporating vapor is governed by the iso-

topic composition of the water, ambient vapor isotopic composition, exchange and mixing processes at the water-air interface as well as the local meteorological conditions. These controlling parameters were considered separately or simultaneously for explaining the observation best quantifying the evaporation mechanism adopted in the Craig-Gordon models. The Traditional Craig-Gordon (Craig and Gordon, 1965) (TCG) and the Unified Craig-Gordon (UCG) (Gonfiantini et al., 2018) equations were used to predict the isotopic composition of evaporation flux after incorporating different molecular diffusivity ratios at varying fractions of molecular and turbulent diffusion. The best match for between the modelled and observed values is seen by using the MJ and CD molecular diffusivity ratios whereas the largest mismatch is for the PW values of the molecular diffusivities. The results ascertain the importance of the fraction of molecular vs turbulent fraction (i.e. isotopically fractionating vs non fractionating exchange) used to predict the isotopic composition of the evaporation flux in these Craig-Gordon models. $UCG_{x=0.8}^{MJ}, UCG_{x=0.6}^{CD}, TCG_{x=0.6}^{MJ}$ and $TCG_{x=0.7}^{CD}$ models predicted the slope and the intercepts of dxs vs meteorological parameters with an appreciable accuracy and consistent with the observations. The remaining difference between the observed and simulated isotopic composition of water vapor is explained by incorporating an advective framework where the advected vapor mass is assumed to mix with the locally generated vapor in a mixing model. The assignment of the advective component is based on the path followed by the air-masses calculated by the HYSPLIT trajectory model. The relative contribution of advected and locally evaporated fluxes was estimated by assigning end member isotopic composition and solving in a two-component mixing framework. The approximation of the locally generated end member composition is based on $UCG_{x=0.8}^{MJ}, UCG_{x=0.6}^{CD}, TCG_{x=0.6}^{MJ}$ and $TCG_{x=0.7}^{CD}$. The advected moisture flux is assigned values based on the origin and path followed by the back trajectories. It is found that beyond 65°S latitude lighter isotope values observed in the water can be explained by invoking mixing of Antarctic vapor with its contribution linearly increasing towards the coast.

Although the advective model can explain the water vapor composition along the transect, nonetheless there can be other sources of advective humidity to the atmospheric boundary layer such as from upper atmospheric layers with different properties, vapor generated from the re-evaporation of rainfall, evaporation of sea spray or sublimation of snow and ice. These processes may occur under conditions which are not possible to take into account due the cryogenic sampling method for collection of water vapor used in this study. The study can be improved and by measuring the water vapor isotopic composition continuously along the transect using a infrared laser spectrometer and conducting a high resolution precipitation sampling during the passage of extra-tropical cyclones.

**Author contributions**

SSD and PG conceptualized the study. SSD and AS performed the sample collection and analysis. SSD wrote the paper and PG supervised the study. AK provided the resources during the expeditions.

## Acknowledgments

We thank the Ministry of Earth Sciences, Government of India, for financially supporting the Southern Ocean Expeditions, under which the present work was carried out. We also thank the National Centre for Polar and Ocean Research, Goa, India for providing the necessary support. The scientists, Sarat Chandra Tripathy, Rajani Kanta Mishra, Anoop Mahajan, Bhaskar Parli Venkateswaran and members of the Southern Ocean expeditions with whose cooperation the meteorological instruments were maintained and measurements performed. Sabu Prabhakaran for providing the salinity data. We finally thank the two anonymous referees for their comments and suggestions which greatly improved the quality of this manuscript.

### Data Availability

The data that support the findings of this study have been uploaded as a supplementary document.

### Declaration of interests

The authors declare that they have no known competing financial interests or personal relationships that could have appeared to influence the work reported in this paper.

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

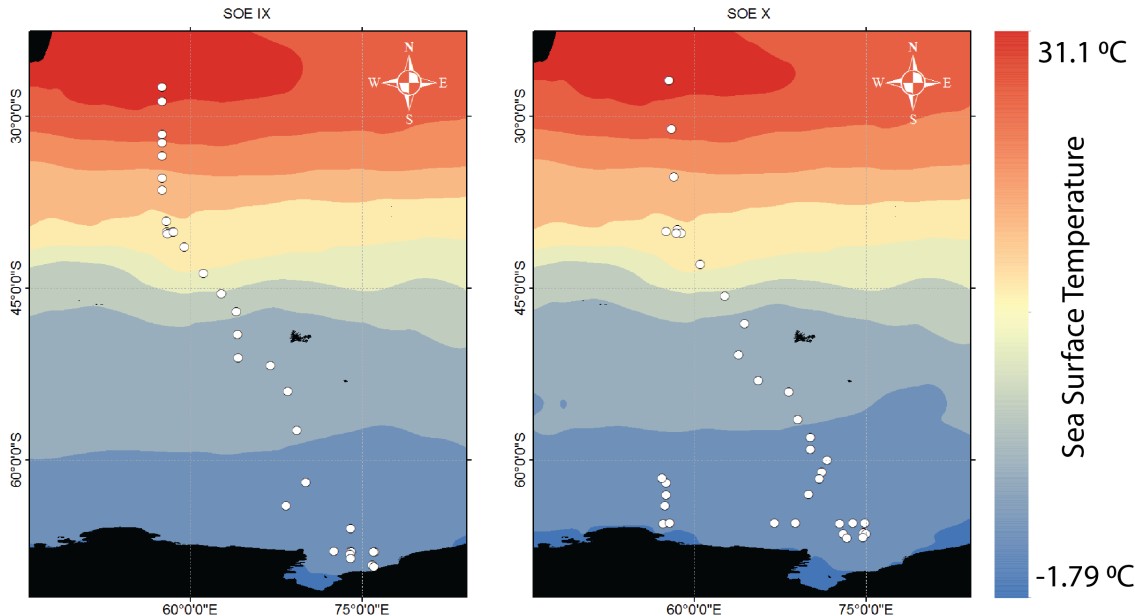

**Figure 1.** The water vapor sampling locations during the two expeditions (January 2017 (SOE-IX) and December 2017 to January 2018 (SOE-X)) shown as open circles overlain on the map of mean monthly sea surface temperature during the two expeditions. The sea surface temperature data is from Reanalysis dataset Kanamitsu et al. (2002).

**Table 1.** Descriptive statistics of the water vapor isotopic composition.

| | $\delta^{18}O(‰)$ | $\delta^{2}H(‰)$ | **d-excess**$(‰)$ |
|---|---|---|---|
| SOE IX Water Vapor(n=34) | | | |
| *Max* | -10.86 | -80.79 | 18.65 |
| *Min* | -27.47 | -221.38 | -8.37 |
| *Mean(Stdev)* | -16.96($\pm$5.25) | -130.35($\pm$44.43) | 5.35($\pm$8.06) |
| SOE X Water Vapor(n=37) | | | |
| *Max* | -11.46 | -88.03 | 14.54 |
| *Min* | -21.18 | -163.28 | -23.71 |
| *Mean(Stdev)* | -15.77($\pm$2.53) | -126.07($\pm$20.23) | 0.08($\pm$8.46) |

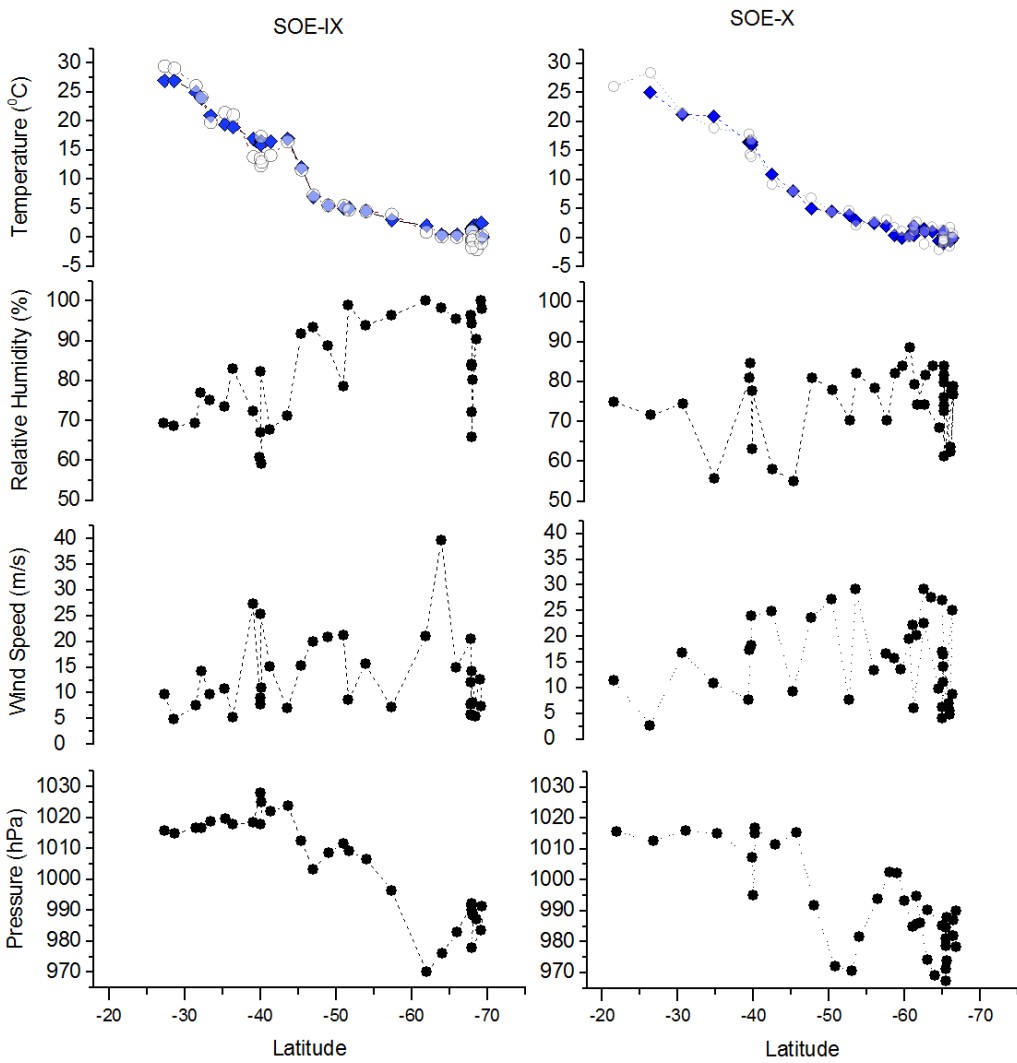

**Figure 2.** Latitudinal variability of measured meteorological parameters, temperature, relative humidity, wind speed and atmospheric pressure. Filled blue diamonds and open circles in the temperature plot represent the sea surface temperature and air temperature respectively.

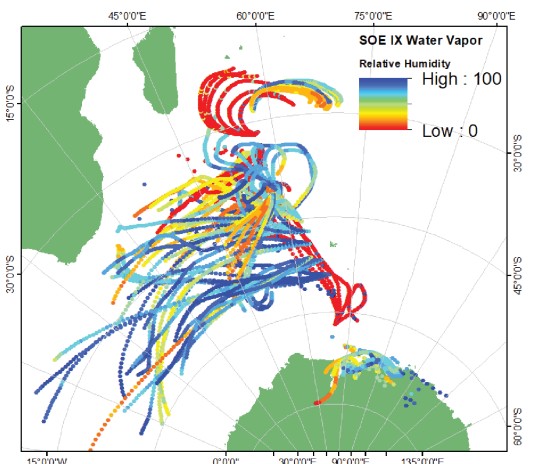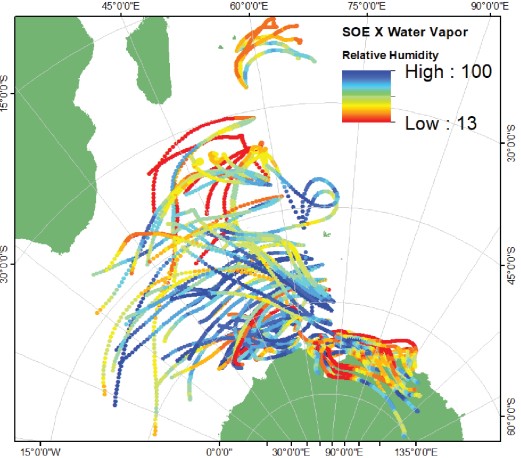

**Figure 3.** 72 hours back trajectories calculated using HYSPLIT with Reanalysis data as forcing. The trajectories shown are for three heights surface, 500m and 1500m above the mean sea level and the colors depict the variation of relative humidity along the trajectories.

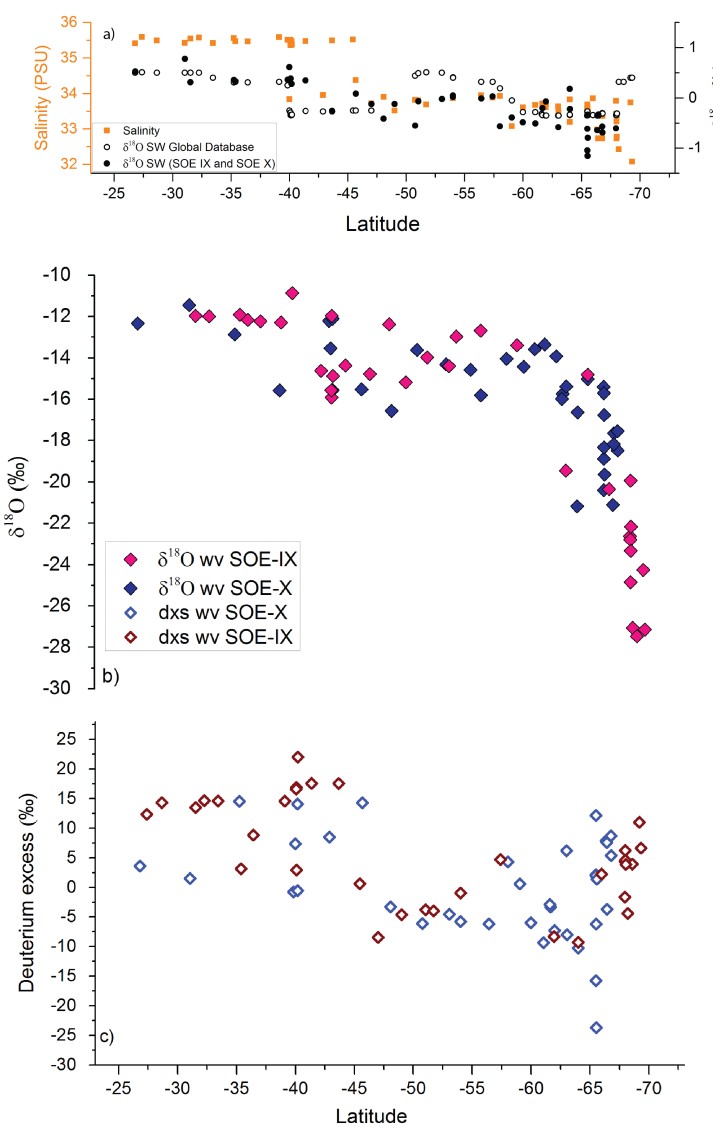

**Figure 4.** a) Measured $\delta^{18}O_{SW}$ as black filled circles and values of surface water isotopic composition extracted from the global sea water $\delta^{18}O_{SW}$ database along the latitudinal transect (open black circles). Also plotted as orange filled squares are the salinity values along the transect b) Pink and purple filled diamonds depict the $\delta^{18}O_{WV}$ of water vapor samples collected during the SOE-IX and SOE-X respectively at height of $\approx$ 15m above the water surface. c) latitudinal variation of dxs in water vapor samples shown as open red and purple diamonds for SOE-IX and SOE-X respectively.

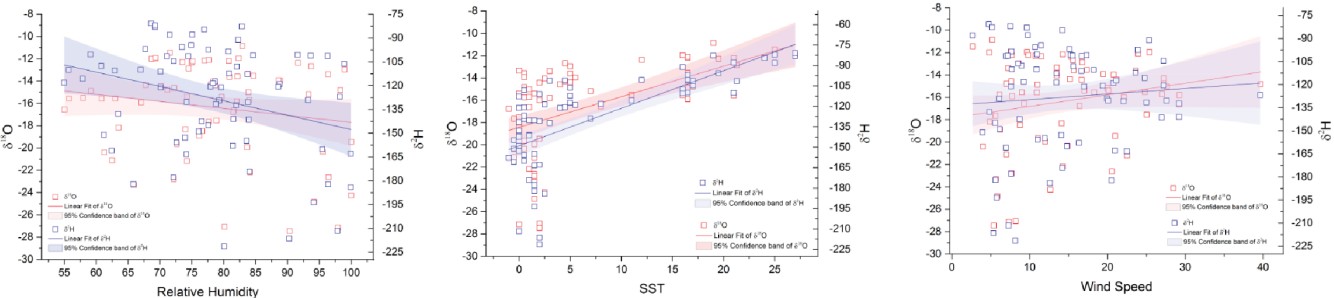

**Figure 5.** Linear regression for isotopic composition of water vapor and physical parameters(sea surface temperature, relative humidity and wind speed). Hollow red and blue squares represent the $\delta^{18}O$ and $\delta^2H$ respectively and the shaded areas depict the 95% confidence bands. The linear regression lines are shown as blue and red for $\delta^2H$ and $\delta^{18}O$ respectively. The slope and intercept of the linear regression equations along with data from Uemura et al. (2008) are listed in Table 1.(S) and Table 2.(S).

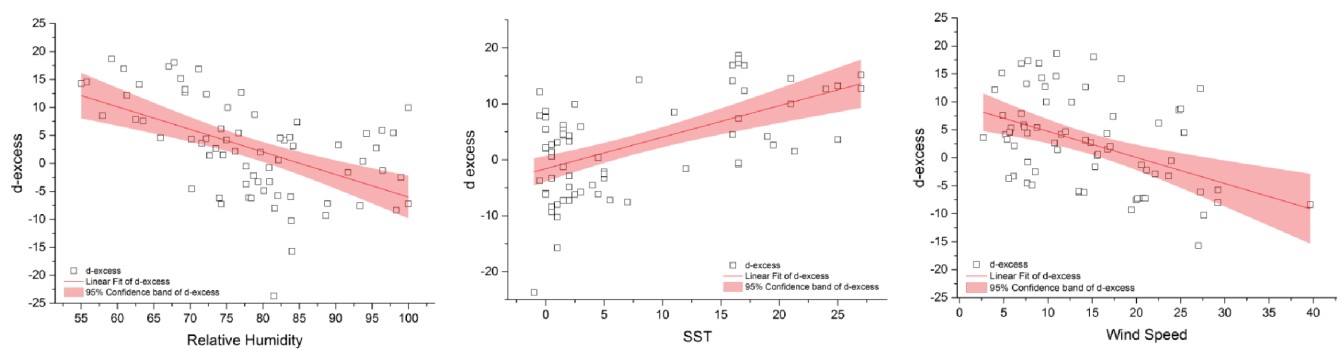

**Figure 6.** Regression plots for d-excess (hollow black squares) in water vapor and the meteorological conditions (relative humidity, sea surface temperature and wind speed). The shaded region depicts the 95% confidence bands of d-excess. The slope and intercept of the regression equations along with data from Uemura et al. (2008) are listed in Table 2.

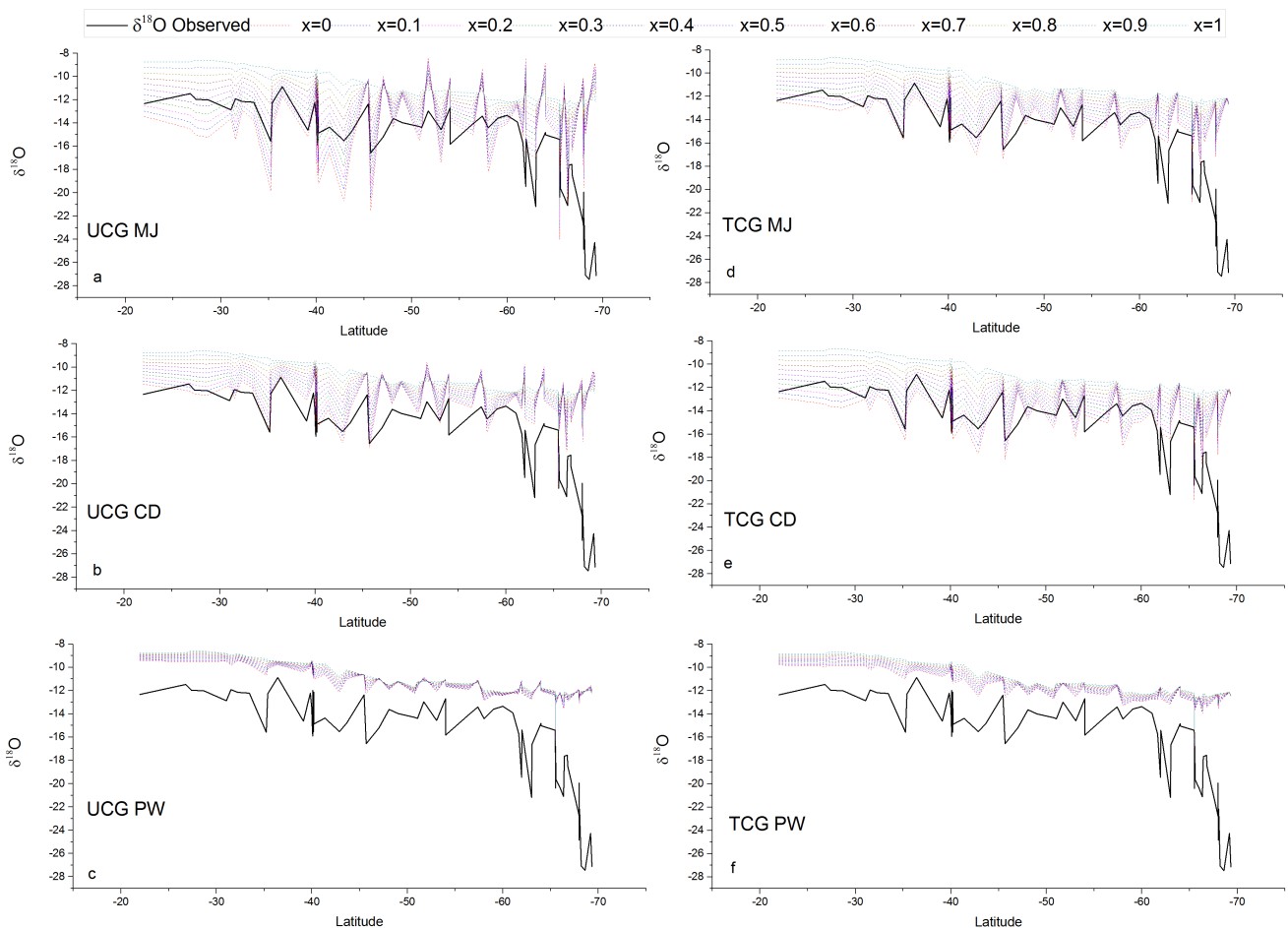

**Figure 7.** Comparison between the latitudinal distribution of the measured water vapor $\delta^{18}O$ (black lines) and that predicted by the TCG and UCG models, employing the global closure assumption for different molecular diffusivity ratios and turbulence indices, shown as colored lines.

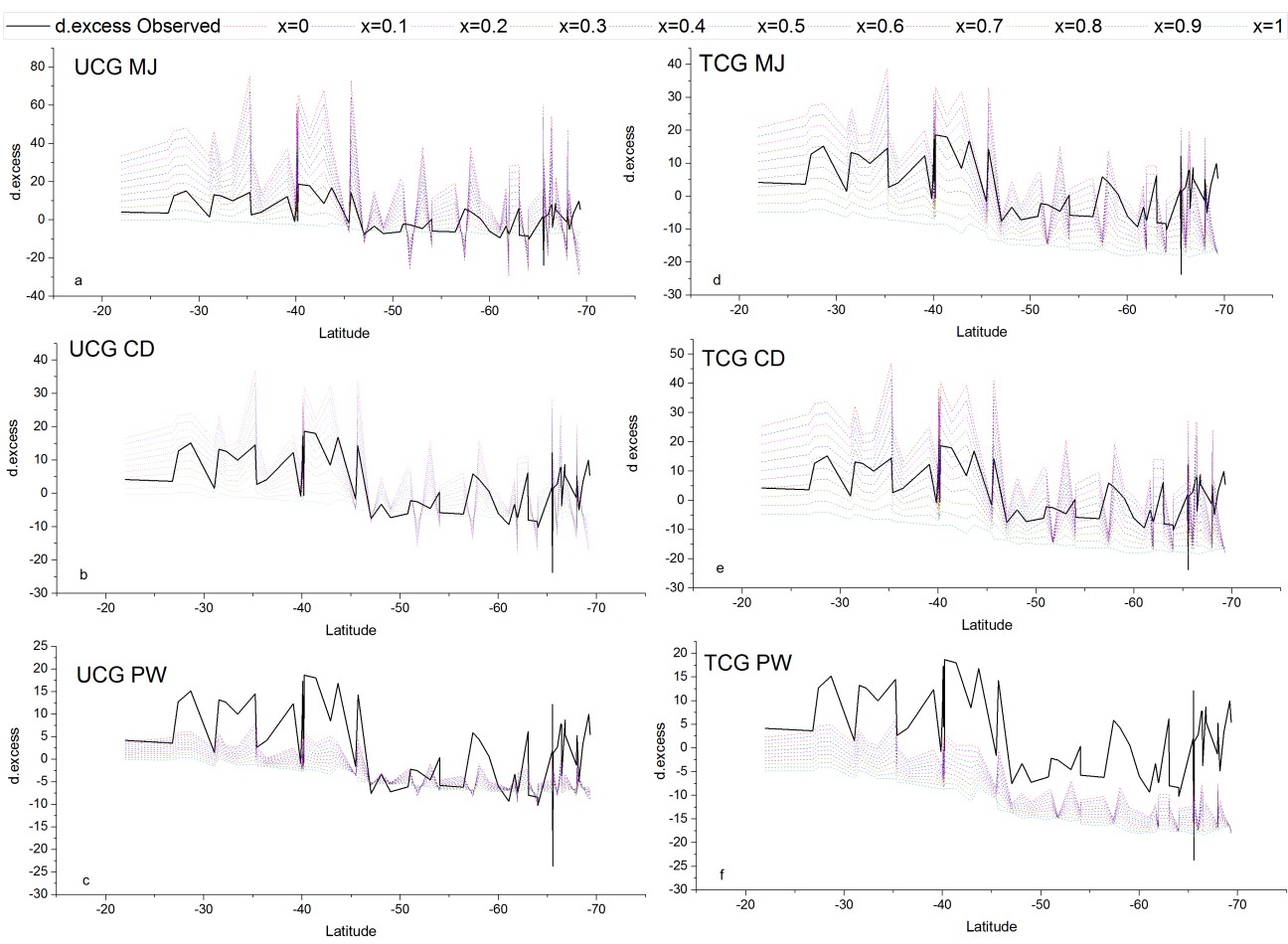

**Figure 8.** Comparison between the latitudinal distribution of the measured d-excess in water vapor (black lines) and that predicted by the TCG and UCG models, employing the global closure assumption for different molecular diffusivity ratios and turbulence indices shown as colored lines.

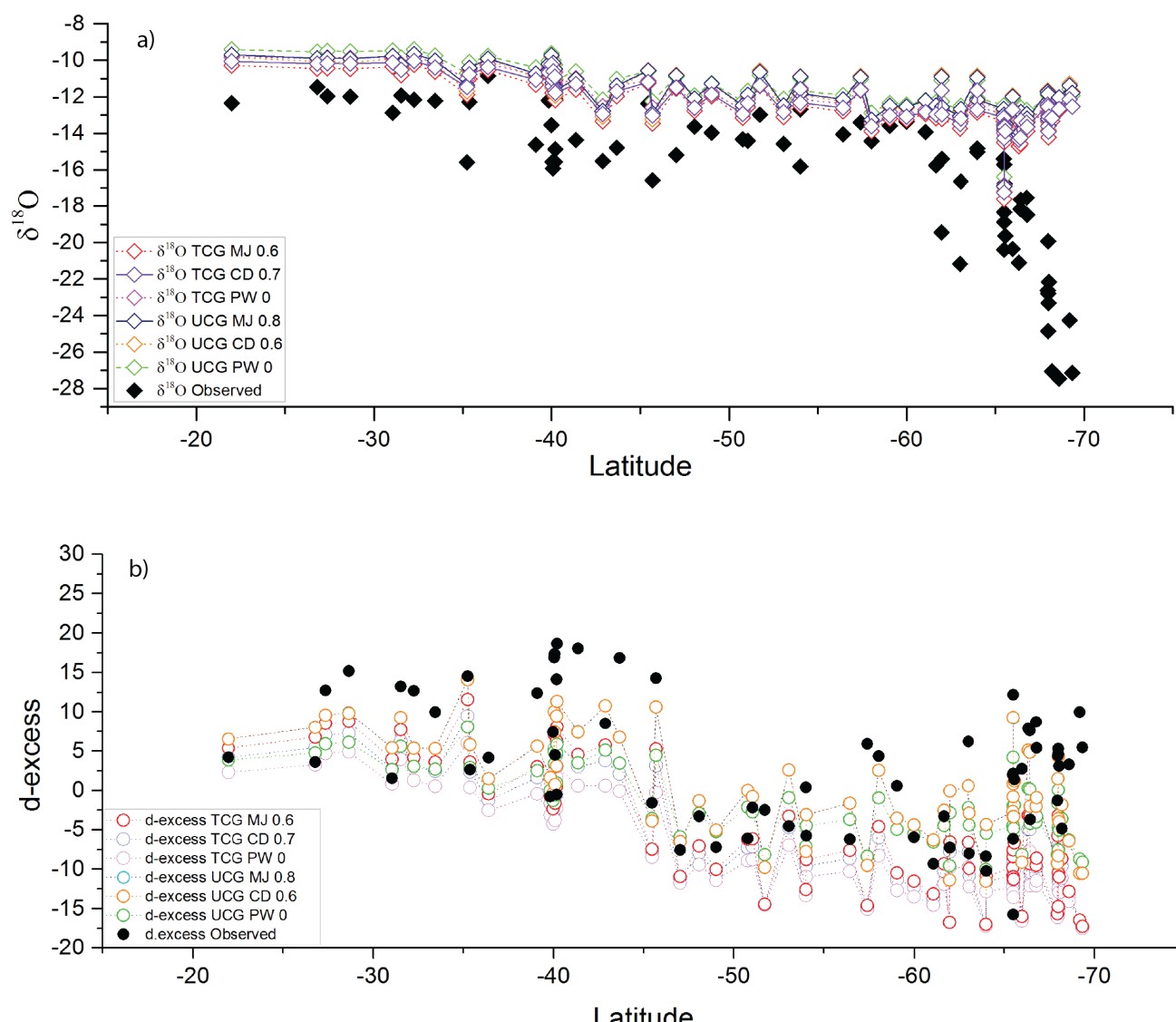

**Figure 9.** Latitudinal variation of the observed $\delta^{18}O$ (a) as filled black diamonds and d-excess (b) as filled black circles and the modelled values (colored open diamonds and circles) for the model runs where the observed slope is comparable to the modelled slope. The statistical parameters analysis of the observed and modelled regression are listed in Table 3.

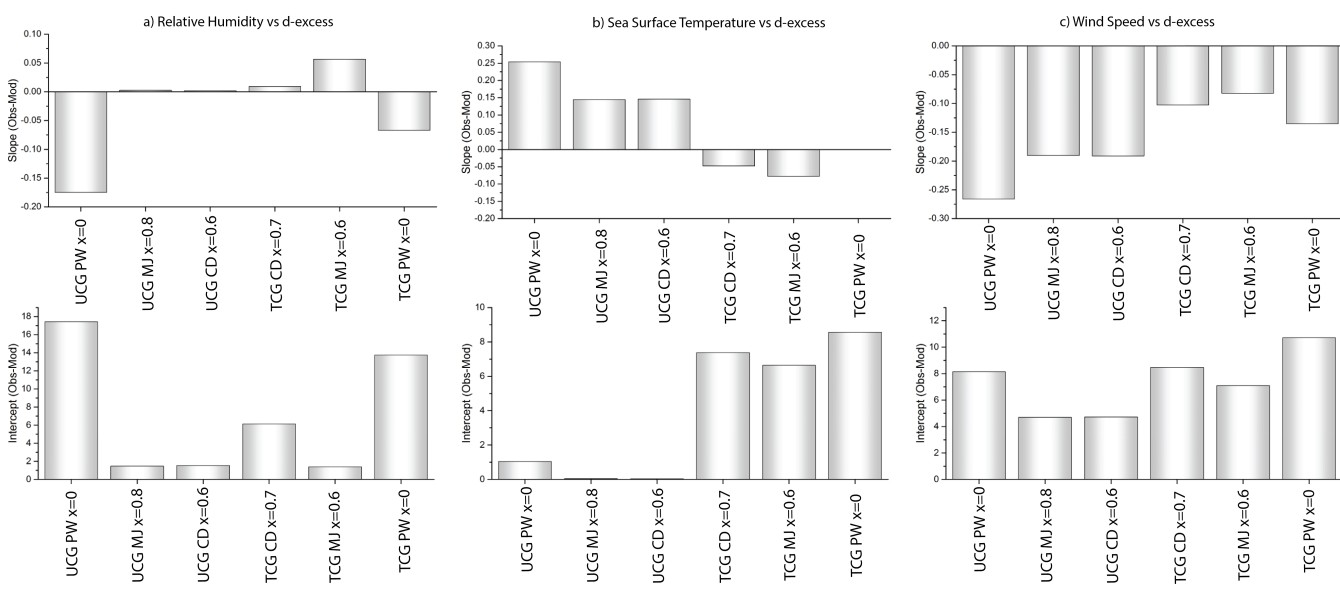

**Figure 10.** Differences between observed and predicted slopes and intercepts of the relationships between d-excess vs relative humidity, sea surface temperature and wind speed.

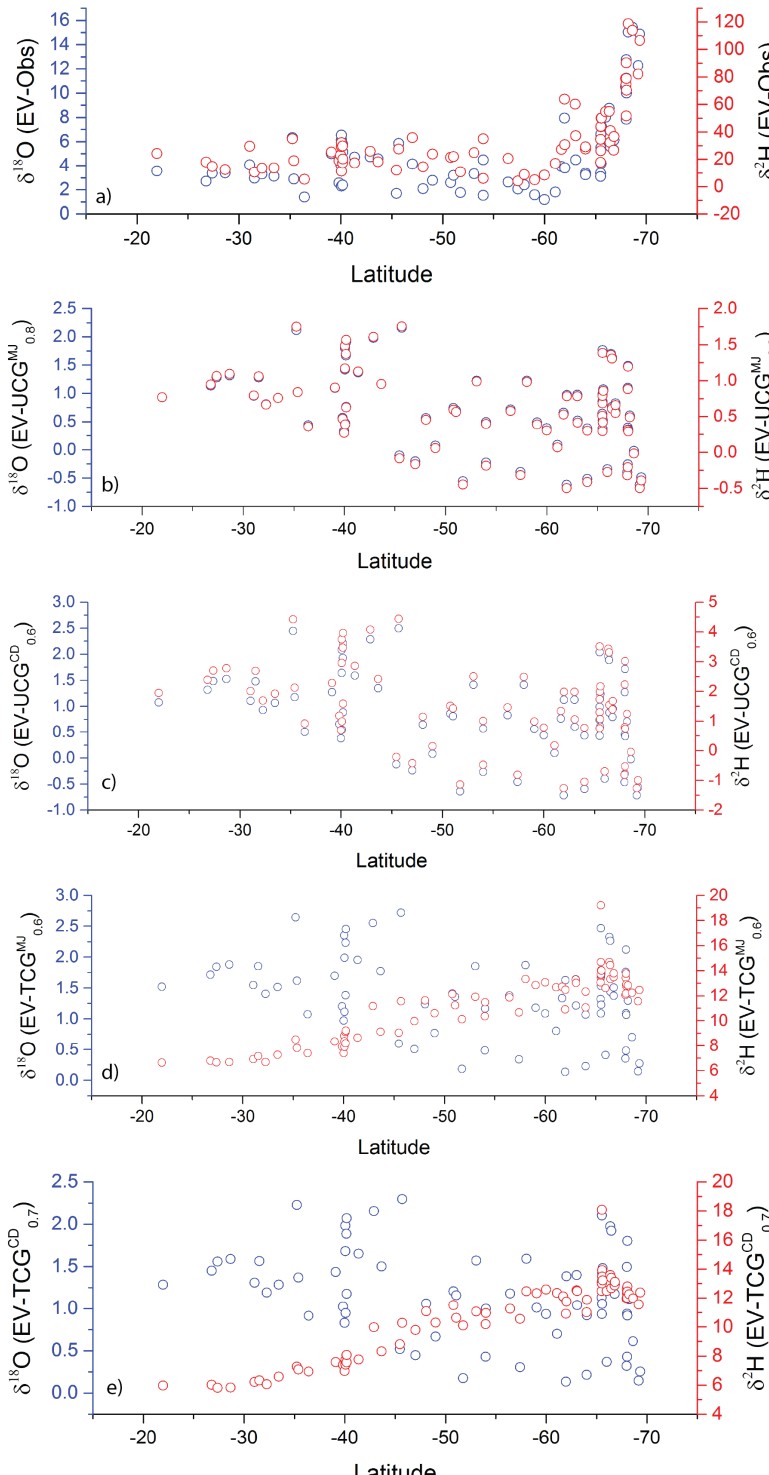

**Figure 11.** a) the difference between the $\delta^{18}O$ (blue columns) and $\delta^2H$ (red open circles) of equilibrium vapor and observed water vapor isotopic composition. b-e) shows difference between the $\delta^{18}O$ and $\delta^2H$ equilibrium vapor and that predicted by the best fit model runs.

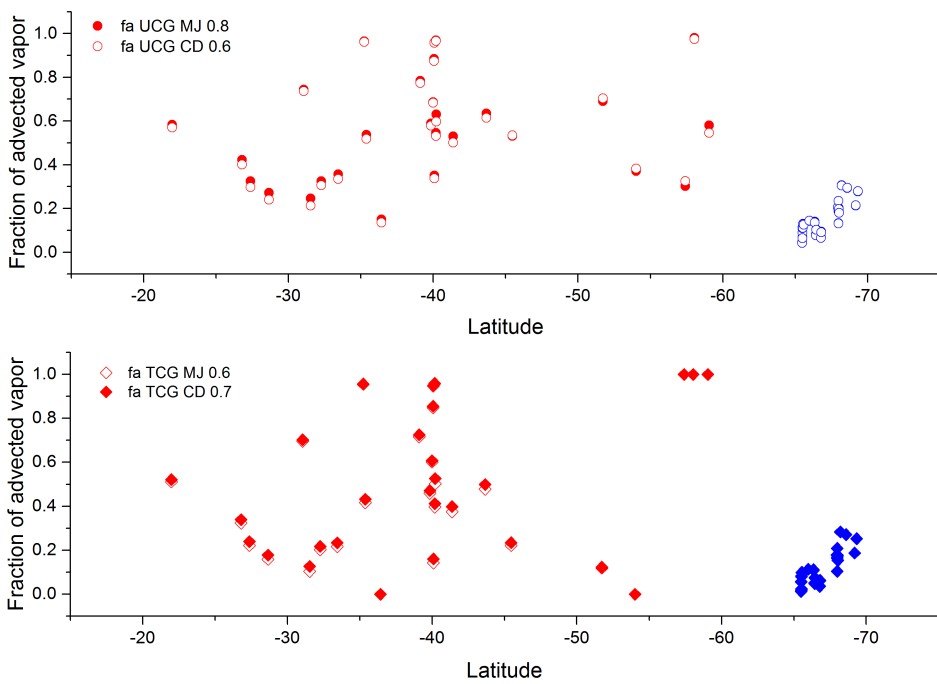

**Figure 12.** a) Fraction of advected vapor that explains the water vapor isotopic composition for the best fit model runs. Red and blue colors depict the different end member compositions used for calculations.

**Table 2.** Slope, intercept and $r^2$ of the linear regression equations between meteorological parameters (relative humidity, sea surface temperature and winds speed) and d-excess for different sample classifications. Also listed are the regression parameters for the data from Uemura et al. (2008).

| Met. vs d-excess | Classification | Intercept | | Slope | | Statistics |
| --- | --- | --- | --- | --- | --- | --- |
| | | Value | Standard Error | Value | Standard Error | R-Square(COD) |
| Relative Humidity | ALL | 34.31 | 6.23 | -0.40 | 0.08 | 0.28 |
| | ALL North of 65°S | 46.36 | 6.57 | -0.56 | 0.08 | 0.49 |
| | ALL South of 65°S | 8.35 | 12.49 | -0.08 | 0.15 | 0.01 |
| | SOE IX North of 65°S | 57.40 | 6.15 | -0.64 | 0.08 | 0.77 |
| | SOE X North of 65°S | 48.66 | 8.28 | -0.64 | 0.11 | 0.61 |
| | ALL SOE X | 53.37 | 8.93 | -0.71 | 0.12 | 0.51 |
| | ALL SOE IX | 42.72 | 6.54 | -0.45 | 0.08 | 0.51 |
| | Uemura All | 54.12 | 4.27 | -0.58 | 0.05 | 0.66 |
| | Uemura North of 65°S | 55.71 | 5.82 | -0.61 | 0.08 | 0.62 |
| Sea Surface Temperature | ALL | -1.58 | 1.15 | 0.56 | 0.10 | 0.31 |
| | ALL North of 65°S | -4.83 | 1.46 | 0.74 | 0.11 | 0.52 |
| | ALL South of 65°S | 0.59 | 2.06 | 1.50 | 1.81 | 0.03 |
| | SOE IX North of 65°S | -5.54 | 2.63 | 0.84 | 0.16 | 0.56 |
| | SOE X North of 65°S | -4.18 | 1.76 | 0.56 | 0.16 | 0.35 |
| | ALL SOE X | -2.19 | 1.62 | 0.43 | 0.18 | 0.14 |
| | ALL SOE IX | -0.36 | 1.60 | 0.58 | 0.12 | 0.42 |
| | Uemura All | 4.13 | 0.98 | 0.79 | 0.12 | 0.43 |
| | Uemura North of 65°S | 3.43 | 1.35 | 0.85 | 0.13 | 0.53 |
| Wind Speed | ALL | 9.40 | 1.97 | -0.47 | 0.12 | 0.18 |
| | ALL North of 65°S | 11.68 | 2.54 | -0.53 | 0.14 | 0.24 |
| | ALL South of 65°S | 7.74 | 3.22 | -0.55 | 0.25 | 0.19 |
| | SOE IX North of 65°S | 15.16 | 3.35 | -0.61 | 0.20 | 0.31 |
| | SOE X North of 65°S | 5.93 | 3.67 | -0.33 | 0.19 | 0.11 |
| | ALL SOE X | 6.08 | 2.96 | -0.38 | 0.17 | 0.13 |
| | ALL SOE IX | 11.58 | 2.51 | -0.47 | 0.17 | 0.20 |

**Table 3.** Slope, intercept and $r^2$ of the linear regression equations between observed and modelled $\delta^{18}O$ and d-excess for the best fit models for samples collected north of 65°S.

| Observed vs Modelled | | Intercept | | Slope | | Statistics | |
|---|---|---|---|---|---|---|---|
| | | Value | Standard Error | Value | Standard Error | Adj. R-Square | Root-MSE (SD) |
| $\delta^{18}O$ All | UCG MJ 0.8 | -8.88 | 0.57 | 0.18 | 0.03 | 0.28 | 1.15 |
| | UCG CD 0.6 | -9.06 | 0.59 | 0.17 | 0.04 | 0.26 | 1.20 |
| | TCG MJ 0.6 | -9.42 | 0.57 | 0.19 | 0.03 | 0.30 | 1.15 |
| | TCG CD 0.7 | -9.15 | 0.55 | 0.19 | 0.03 | 0.32 | 1.12 |
| $\delta^{18}O$ North of 65°S | UCG MJ 0.8 | -6.45 | 0.89 | 0.34 | 0.06 | 0.38 | 0.84 |
| | UCG CD 0.6 | -6.50 | 0.91 | 0.34 | 0.06 | 0.38 | 0.86 |
| | TCG MJ 0.6 | -7.02 | 0.91 | 0.34 | 0.06 | 0.38 | 0.86 |
| | TCG CD 0.7 | -6.89 | 0.91 | 0.34 | 0.06 | 0.37 | 0.86 |
| d-excess All | UCG MJ 0.8 | -0.94 | 0.64 | 0.47 | 0.07 | 0.39 | 5.08 |
| | UCG CD 0.6 | -0.94 | 0.64 | 0.47 | 0.07 | 0.39 | 5.07 |
| | TCG MJ 0.6 | -6.46 | 0.75 | 0.58 | 0.08 | 0.41 | 6.00 |
| | TCG CD 0.7 | -7.39 | 0.71 | 0.55 | 0.08 | 0.41 | 5.66 |
| d-excess North of 65°S | UCG MJ 0.8 | -0.35 | 0.63 | 0.60 | 0.07 | 0.63 | 4.07 |
| | UCG CD 0.6 | -0.36 | 0.63 | 0.60 | 0.07 | 0.63 | 4.06 |
| | TCG MJ 0.6 | -4.93 | 0.72 | 0.74 | 0.08 | 0.67 | 4.67 |
| | TCG CD 0.7 | -5.85 | 0.68 | 0.70 | 0.07 | 0.66 | 4.41 |