# Peer review of "Craig-Gordon model validation using stable isotope ratios in water vapor over the Southern Ocean"

_Atmospheric Chemistry and Physics, 2019_

## Referee Comment (RC1) · Anonymous Referee #1 · 21 Feb 2020

The paper presents isotopic water vapor and meteorological data collected during two 'summer' cruises on RV S.A. Agulhas in 2017-2018 is the Indian Ocean sector of the southern Ocean. The sample data are not numerous, but are very valuable, as meteorological conditions are well described and cover a large spread of conditions. The paper attempts two things: first to check the local equilibrium assumption that atmospheric water vapor results from local sea water evaporation using two models (TCG and UCG), and then attempts to find where and why this breaks down (in the southern part with continental air outflow).

[Figure]

The approach is valuable, but should be much more clearly outlined. For example, lines 40-45 of the introduction should be expanded and it should be made clearer what is the approach adopted. This should be reminded at the beginning of the discussion 4 (line 143), so that the reader does not have to wait on lines 219-220 to understand what is the approach followed (first 'local' evaporation, thus the equilibrium model; then, remote/mixing contributions).

When assuming local evaporation, the abstract and conclusion state that the UCG model with a 0.5 CD molecular diffusivity ratio performs best. I find that conclusion not substantiated, and believe that there are cases with TCG which perform as well. This could be clarified.

Also, note that this issue of non-local contributions is also addressed experimentally for different weather condition in Benetti, M., J.-L. Lacour, A.E. Sveinbjörnsdottir, G. Aloisi, G. Reverdin, C. Risi, A.J. Peters, H.-C. Steen-Larsen. A framework to study mixing processes in the marine boundary layer using water vapor isotope measurements. Geophys. Res. Lett., published 10 March 2018 https://doi.org/10.1002/2018GL077167

The main issues I have are the followings: 1: how well were meteorological parameters measured (in particular, for relative humidity, and was this measurement done very close to where the air was collected (upper level). I find the very large relative humidities in the southern part of the first cruise transect surprising in view of the much weaker relative humidities of the second cruise. Of course, this is possible, but would suggest, almost fog-like conditions for the first cruise. If this is the case, this could involve very different processes and thus expected isotopic properties of the near-surface air mass (what type of clouds, then; subsidence or not?)

2: The meteorological situations are described in ways which are too vague. For example, names of regions a little strange on lines 82-86 (for example melting/freezing starting at 47°S?) On what is it based? What does it represent? I think that the different situations: cyclonic/anticyclonic/precipitating systems, presence of rain or snow
should all be also reported and analyzed to provide a more relevant context that just the HYPSLIT trajectories, which as they are triplets are very hard to read as presented (and are not really commented upon. . .). Also, in which conditions were the high-winds encountered

3: I was surprised to find why the (local equilibrium) model fit seem to work less well with the near surface samples. Any explanation for that? Alltogether, how samples are collected at the two levels should be more clearly described (maybe in supplementary materials, see comments below). Otherwise, it is very hard to understand what to make of the differences in results for the two sets of data.

I will now provide a few comments on the figures: Figure 1 : What are the white circles (compared to all the other red circles) ?

Caption figure 2, last sentence: 'Read and blue colors. . . temperature, respectively.' It is hard to differentiate the two colors. Maybe air temperature should be open dots with no color inside?

Figure 2 suggests that average conditions were with much less relative humidity at high latitudes during second cruise than during the first one. What is the big difference between the two cruises. I find relative humidity so high and for so long on SOE-IX (first cruise) a bit surprising. In particular, this does not seem to be so substantiated by figure 3, but I don't read very well figure 3, which I don't find very clear. It does not seem either the values that are retained afterwards for humidity (such as in figure 6). Have they been adjusted afterwards?

Figure 4, top panel a) suggest south of 40°S SOE-X lower than SOE-IX, itself lower than SW global database. It is a bit surprising, but could be associated with different surface salinities and freshwater sources. What are the salinities and how do they compare with surface water isotopic composition; differences between the two cruises? For figure 4, b), d18O SOE-IX and SOE-X seem rather similar, except maybe some d18O Swv de SOE-X, which could be a little higher near 40-50°S. For dxs less obvious. I

[Figure]

don't see a clear (panel c) difference in dxs near 45-65°S between the two cruises, despite very different relative humidities? (big apparent jump near 45°S). Last sentence of the caption of Figure 4: This is not just zonal variation, but maybe statistical distributions grouped by latitudinal bands.

Figure 5: dependency in humidity seems dubious, but could it be poor measurements and would'nt it be then important to separate the cruises (issue also of cross wind-dependency) Water samples south of 65°S clearer, as well as Swv wamples (but very scattered and how does one measure those). Plot on b) dependency of SST is clearly wrong panel (not right SST! And same clearly as for a; this raises issues of whether one trusts fully the other figures).

Figure 6 on d-excess dependency and meteorological conditions seem rather in agreement (north of 65°S) with Uemura et al. (2008) (also a bit shifted down compared to Uemura's values which could suggest an average bias in one of the data sets).

Figure 7. UCG presents some strange peaks. Why? That could be an argument for preferring TCG, but I don't know how realistic the different ocean surface conditions selected are. On this figure, there is no particular need for the colors of the diamonds

Figure 8: interesting. Actually, both UCG and TCG models don't explain observed d-excess at humidity larger than 90%. Otherwise, dependency in SST and wind speed seem OK for both (TCG gives a reduced range compared to UCG, so should the results be considered better?). Some cases better both in term of correlation and small misfits in slope and intercept (figure 9; the caption should mention whether the comparisons are done with UCG or TCG). The different curves and what they mean (their caption) are hard to read on figures 7 to 10.

Figure 10, probably interesting, but very hard to understand what is presented. Caption should be clearer, and as is does not explain what is presented. The yellow and green bars are too difficult to separate, and should be presented separately (for example one above the other)

In Supplementary material, presenting d18Osw is instructive, but it would be good to add a salinity column to increase the possible use of these data (the data were collected from a rosette with a Seabird CTD, so salinity was measured). It can also help validity-controlling the isotopic sea water data. For example, there is an isolated very negative value: if salinity low and/or collected near an iceberg, it is possible. Otherwise, questionable.

Detailed (mostly minor or editorial) comments: Line 53: '. . . shows the sampling locations. ' How is Swv collected: at which height above the sea surface? How is spray avoided whether it is for Swv or Nwv. . .) .The detail of the collection method could be key for the results obtained. Later, it is mentioned that Swv samples could only be collected by fair weather. What kind of wind/swells/sea states are conductive to this measurement. All these relevant informations should be provided in the supplementary document.

l. 72 '. . ., the change between trajectories corresponding to trade winds from the ones corresponding to westerlies happened at 31°S, whereas at 630S, a change. . .'

l. 106: the sentence 'The role of sst in governing d-excess. . .' This is only direct role, there is the indirect one of the air water mass and dependency on temperature.

l. 113: 'The strength of the correlation is slightly higher. . .'

l. 120: more sensitivity of ïĄď'H to wind speed is not what comes out clearly from figure 5b. It might be a scaling issue of the ïĄď'18O versus ïĄď'D (note that R-square are small for both variables)

Line 129, the slope is the opposite in this latitude range between the two expeditions. This illustrates, I believe, either very different weather conditions (as seen in r of figure 2) or some issue somewhere, which would need to be further clarified Line

132? Sentence?

l. 134: '. . . complement. . .'

Line 135: correlation with SST of d18OSwv. Interesting, but these samples only taken when sea state not strong. Relating the dxs of seawater with relative humidity is a bit strange, as this is close to sea surface but with little wind/sea. I am not sure of what that brought?

l. 140: '. . . to the near-surface water vapour. . .' I am wondering whether the regression is significant, and in that case, whether this sentence is not anecdotical, and should be omitted.

In description of UCG (l. 157-191), the ratio term is ïĄǧ-h, which makes sense in this context. However, note that at saturation (fog, for example), there is a definition issue. For that, The near saturation values reported on Figure 2 almost everywhere south of 50°S during SOE-IX is a major issue. How was this dealt with (and again, what confidence does one have in these near-saturated conditions, not witnessed the second year).

Line 175: remove a 'same'. After that, the global closure is assumed as in Merlivat (1978)

l. 201. I would mention the caveat of the issue of sea spray for high winds. Evaporation from sea spray is a large contribution to total evaporation in these conditions, and follows different laws (in the extreme case of all sea spray evaporating, this would for example yield Rev=RI) There is also the caveat of below freezing temperature, and low SST close to sea ice formation temperature (just below fresh water freezing points), but I gather from figure 2 that this almost did not happen (correct?)

l. 219-220. This sentence is key. I think that this should be presented earlier.

l. 229: '. . . where westerlies dominate. . .'

L. 233: end of sentence missing

l. 236 '. . . is less, and is largely local. . .'

---

## Referee Comment (RC2) · Anonymous Referee #2 · 2 Mar 2020

The submitted paper by Dar et al presents an analysis of isotopic data from southern latitudes. These data are compared with a set of models and associated coefficients. While worthwhile, a revision of the structure and error analysis of conducted here would greatly improve this submission. First, the overall structure of this paper is a bit jumbled. Much of the discussion material about the CG models should be moved into the methods section. Similarly, the discussion of how the HYSPLIT back trajectories were ran should also be described in the methods section. Second, the error analysis needs to be more fully documented. The testing of different model formulations and

parameters is helpful, however the authors do not fully evacuate the errors and biases associated with each model. A more rigorous description of errors across all variables is needed. Finally, is there an optimal set of an parameters that others should use (i.e. what value of x and fractionation factors minimizes errors and bias)?

L4: Add latitudes numbers here. L5: Reword the sentence that starts "The inter annual", its not clear what your trying to say L19: Nearly and your double tilde are redundant. L25: Missing an 'A' at the beginning? L52: Define what you mean here by boundary later? Where these really at the boundary layer? L65: This text on HYSPLIT methods should move to section 2. L76: Careful with your terminology here. A positive delta value signifies that it was more enriched in heavy isotopes relative to VSMOW only. L110-L120 What are the significance and or uncertaintie3s of these regressions. L132: Why not put the regression coefficients and stats from Figs 5 & 6 in a table L143-201: This needs to all move to methods. L197: Please directly state the numbers you used here for the diffusivities L209: While differences between the slopes and intercepts are interesting, a more error though analysis should be conducted. What is the overall bias associated with each model, what are the root mean squared errors, etc.
* * *

---

## Author Comment (AC1) · 28 May 2020

We appreciate reviewer's comments.

Below we have compiled our answers to the reviewer's questions and comments.

Please note that the Swv samples were excluded in the modified manuscript as lack meteorological parameters for them. Therefore only samples Nwv at the sampling height of 15m were taken for the paper.

All the modified and new figures and tables have been included in the Author Comments.

The reviewer's comments and questions are presented as italic while our answers are written as bold.

*The paper presents isotopic water vapor and meteorological data collected during two 'summer' cruises on RV S.A. Agulhas in 2017-2018 is the Indian Ocean sector of the southern Ocean. The sample data are not numerous, but are very valuable, as meteorological conditions are well described and cover a large spread of conditions. The paper attempts two things: first to check the local equilibrium assumption that atmospheric water vapor results from local sea water evaporation using two models (TCG and UCG), and then attempts to find where and why this breaks down (in the southern part with continental air outflow). The approach is valuable, but should be much more clearly outlined. For example, lines 40-45 of the introduction should be expanded and it should be made clearer what is the approach adopted. This should be reminded at the beginning of the discussion 4 (line 143), so that the reader does not have to wait on lines 219-220 to understand what is the approach followed (first 'local' evaporation, thus the equilibrium model; then, remote/mixing contributions).*

**The introduction is now modified for clarity with more description on the approach adopted in the study.**

*When assuming local evaporation, the abstract and conclusion state that the UCG model with a 0.5 CD molecular diffusivity ratio performs best. I find that conclusion not substantiated, and believe that there are cases with TCG which perform as well. This could be clarified.*

**In the modified version of this study, the Craig-Gordon models (both UCG and TCG) were run for different fractions of the turbulent indices, x (i.e. the fraction of molecular and turbulent diffusion). The value of x was varied from 0 to 1 with an interval of 0.1. We found that the UCG and TCG models both perform well for MJ and CD molecular diffusivity ratios, while for the PW diffusivity ratios the difference between the observed and the modelled values is the largest. This is explained in the paper. We found that UCG MJ for x=0.8, UCG CD for x=0.6 and TCG MJ for x=0.6 and TCG CD for x=0.7 perform equally well within the uncertainty limits.**

*The main issues I have are the followings:*
*1: how well were meteorological parameters measured (in particular, for relative humidity, and was this measurement done very close to where the air was collected (upper level). I find the very large relative humidities in the southern part of the first cruise transect surprising in view of the much weaker relative humidities of the second cruise. Of course, this is possible, but would suggest, almost fog-like conditions for the first cruise. If this is the case, this could involve very different processes and thus expected isotopic properties of the near-surface air mass (what type of clouds, then; subsidence or not?)*

**The meteorological parameters were measured at the height of ~15 m from sea level, which was the sampling inlet for water vapour isotopic measurements. The relative humidity was estimated at the dry and the wet bulb temperature using the psychrometric charts. The two cruises differs in the latitudinal coverage area. Water vapour sampling in the first cruise was carried out until 69.3°S, whereas for the second cruise, the sampling was conducted until 66.8 °S. We recorded continuous precipitation over a duration between 20/01/2017-25/01/2017 (SOE IX) when relative humidity level approached higher values.**

2. *The meteorological situations are described in ways which are too vague. For example, names of regions a little strange on lines 82-86 (for example melting/freezingstarting at 47 ∘ S?) On what is it based? What does it represent? I think that the different situations: cyclonic/ anticyclonic/ precipitating systems, presence of rain or snow should all be also reported and analyzed to provide a more relevant context that just the HYPSLIT trajectories, which as they are triplets are very hard to read as presented(and are not really commented upon. . .). Also, in which conditions were the high-winds encountered.*

**This section is now modified.**
**The triplet HYSPLIT trajectories are shown together to give comprehensive overview about the overall moisture transport pathways.**
**The Southern Ocean is generally associated with high wind speed conditions.**

3: *I was surprised to find why the (local equilibrium) model fit seem to work less well with the near surface samples. Any explanation for that? Alltogether, how samples are collected at the two levels should be more clearly described (maybe in supplementary materials, see comments below). Otherwise, it is very hard to understand what to make of the differences in results for the two sets of data.*

**The meteorological parameters which goes as input parameters are measured at the height of ~15 m. This would be one of one of the major reason responsible for the discrepancy between near surface samples (Swv) and the samples collected at ~15m above the ocean surface.**
- **Therefore, the Swv samples have been omitted from discussion.**
- **Additional information on the sampling procedures has been added in the supplementary document.**
- **The Swv samples were collected when the ocean surface conditions were calm i.e. when there was no visible wave in the ocean (No wave breaking near the sampling inlet and the ship was kept stationary for conducting the CDT depth profiling and sampling. The ocean beyond 65S latitude was relatively calm as compared to other latitudes.**

*I will now provide a few comments on the figures:*

*Figure 1 : What are the white circles (compared to all the other red circles) ?*
- **The white circles depict the locations of the water vapour (Swv) samples i.e. the vapour samples collected close to the ocean surface by placing the inlet tube just 1 m above the Sea water level. Since these samples were omitted from the discussion now, the figure has been modified and the caption has been rewritten to accommodate the new changes. (Figure 1 in the Author comment file)**

*Caption figure 2, last sentence: 'Read and blue colors. . . temperature, respectively.' It is hard to differentiate the two colors. Maybe air temperature should be open dots with no color inside?*
- **The figure has been modified according to the suggestion. (Figure 2 in the author comment file)**

*Figure 2 suggests that average conditions were with much less relative humidity at high latitudes during second cruise than during the first one. What is the big difference between the two cruises. I find relative humidity so high and for so long on SOE-IX (first cruise) a bit surprising.*

- **There was continuous cloud cover for the days with high RH levels. Also, precipitation event lasted for the entire length with low pressure system as mentioned earlier.**

*In particular, this does not seem to be so substantiated by figure 3, but I don't read very well figure 3, which I don't find very clear.*

- **The triplet HYSPLIT trajectories are shown together to give comprehensive overview about the overall moisture transport pathways. There exists a few cases for both the expeditions when there was a complete mismatch between the observed and the HYSPLIT predicted RH values (governed by the Reanalysis dataset). The RH along the back-trajectories in close to 0 for all elevations. This may be due to missing data values in the input dataset used for generating the HYSPLIT model results.**

*It does not seem either the values that are retained afterwards for humidity (such as in figure 6). Have they been adjusted afterwards?*

- **In the old Figure 6, samples collected exclusively north of 65S latitude were plotted. The reason being the mixing of Antarctic vapor with depleted heavy isotope values south of 65S latitude. Large relative humidity were mostly observed for the region south of 65S and during the passage of an extratropical cylone. This figure has been modified to include all the data points and the regression equations for various sample classifications have been provided in a separate table.**

*Figure 4, top panel a) suggest south of 40 ∘ S SOE-X lower than SOE-IX, itself lower than SW global database. It is a bit surprising, but could be associated with different surface salinities and freshwater sources. What are the salinities and how do they compare with surface water isotopic composition; differences between the two cruises?*

- **The cruises were not during the same month, for SOE IX the sampling was performed in January 2017 while for SOE-X the samples were collected during December-2017 to mid January-2018 . The salinity values measured along the transect are now included in the revision.**

*For figure 4, b), d18O SOE-IX and SOE-X seem rather similar, except maybe some d18O Swv de SOE-X, which could be a little higher near 40-50 ∘ S. For dxs less obvious. I don't see a clear (panel c) difference in dxs near 45-65 ∘ S between the two cruises, despite very different relative humidities? (big apparent jump near 45 ∘ S).*

- **The Swv samples is now removed from the modified figure.**
- **In addition to the local factors, the dxs is also controlled by the mixing of the advected vapor  this is probably the reason.**

*Last sentence of the caption of Figure 4: This is not just zonal variation, but maybe statistical distributions grouped by latitudinal bands.*

- **Latitudinal ranges (Zones) are grouped according to the approximate positioning of the general wind patterns (easterly or westerly) decided based on the HYSPLIT trajectories. The caption has been modified to make the idea clear.**

*Figure 5: dependency in humidity seems dubious, but could it be poor measurements and would'nt it be then important to separate the cruises (issue also of cross wind-dependency) Water samples south of 65 ∘ S clearer, as well as Swv wamples (but very scattered and how does one measure those).*

- **We agree the plot is confusing, the plot has been modified and some information has been provided in the tables (Table 2 for d18O and Table 3 for d2H). The relationships for separate cruises has been added.**
- **The sampling for Swv water vapor samples was totally dependent on the ocean conditions encountered. The dependency might be dubious to some extent as the measurements for relative humidity were done at the height of the Nwv samples (~15m above the sea level).**
- **The Swv samples have not been included in the modified version of the manuscript due to the reasons stated before.**

*Plot on b) dependency of SST is clearly wrong panel (not right SST! And same clearly as for a; this raises issues of whether one trusts fully the other figures).*
- **I don't understand this question. Nevertheless, this figure has been modified and additional information has been added in Table 2 and Table 3.**

*Figure 6 on d-excess dependency and meteorological conditions seem rather in agreement (north of 65 ∘ S) with Uemura et al. (2008) (also a bit shifted down compared to Uemura's values which could suggest an average bias in one of the data sets).*
- **This is due to the fact that low temperatures (both sst and air temperatures) are low beyond 65S and hence limit the evaporation. Moreover, beyond 65S there is advection and mixing of very light Antarctic vapor.**
- **This really answers the doubt you raised and clears it regarding the dubiousness of RH measurements. It has been mentioned in the text that if considered separately the relationship is stronger and -0.64h+ 57.4‰ (r2 =0.77) and -0.64h+48.7 (r 2 =0.61) for SOE IX and SOE X respectively comparable to the that from previous measurements in the Southern Ocean (Uemura et al. 2008) -0.61h+55.71(r2=0.63).**
- **The figure has been modified and the additional information has been provided in a table (Table 4)**

*Figure 7. UCG presents some strange peaks. Why? That could be an argument for preferring TCG, but I don't know how realistic the different ocean surface conditions selected are. On this figure, there is no particular need for the colors of the diamonds*
- **The larger variability for in the isotopic values predicted by the both UCG and TCG models for MJ and CD molecular diffusivity ratios and thus maybe a reason for a better representation of the conditions under which evaporation happens in the open ocean by CG models for these molecular diffusivity ratios. The model runs for different molecular diffusivity ratios have been separated in the modified figure and are much clearer now.**
- **The ocean conditions x (turbulence coefficient) are selected are assuming ratios of molecular (x=1) to turbulent transport (x=0) and x=0.5 for equal contribution by molecular and turbulent diffusion. In the modified version the models are run for x varying from 0-1 with an increment of 0.1.**
- **This figure has been modified and the data are presented separately for d18O and d-excess. (Figure 7 shows the comparision between modelled and observed d18O and Figure 8 between modelled and observed d-excess)**

*Figure 8: interesting. Actually, both UCG and TCG models don't explain observed d-excess at humidity larger than 90%. Otherwise, dependency in SST and wind speed seem OK for both (TCG gives a reduced range compared to UCG, so should the results be considered better?). Some cases better both in term of correlation and small misfits in slope and intercept.*

- **This is a known shortcoming of the model as it doesn't perform well for high RH conditions, where greated influence on the isotopic composition is exerted by the advected vapor.**
- **During the period of sampling, high RH conditions were observed when precipitation occurred and when extratropical cyclones were encountered (as evident from the low pressures).**
- **In the modified version as explained before, it was found that UCG MJ for x=0.8, UCG CD for x=0.6 and TCG MJ for x=0.6 and TCG CD for x=0.7 perform equally well within the uncertainty limits. The comparision between the observed and these models have been added as a separate figure. (Figure 10)**
- **The models that best describe the conditions are selected for which there are least differences between the regression parameters (slope and intercept) of the meteorological parameters (rh, sst and wind speed) vs d-excess of water vapor. ( Figure 9, Figure 11 and Figure 12)**

*(figure 9; the caption should mention whether the comparisons are done with UCG or TCG). The different curves and what they mean (their caption) are hard to read on figures 7 to 10.*
- **The comparisons have been presented for both UCG and TCG models. Since in the modified version, the models have been run for x varying between 0-1 with and increment of 0.1 as opposed to 0,0.5 and 1 in the previous version. This additonal information has been clearly presented in Figure 9, Figure 11 and Figure 12 in the modified version.**

*Figure 10, probably interesting, but very hard to understand what is presented. Caption should be clearer, and as is does not explain what is presented. The yellow and green bars are too difficult to separate, and should be presented separately (for example one above the other)*
- **Noted. The figure has been modified, all the model runs which perform well have been included and the caption rephrased. (Figure 13 and Figure 14 in the revised manuscript)**

*In Supplementary material, presenting d18Osw is instructive, but it would be good to add a salinity column to increase the possible use of these data (the data were collected from a rosette with a Seabird CTD, so salinity was measured). It can also help validity-controlling the isotopic sea water data. For example, there is an isolated very negative value: if salinity low and/or collected near an iceberg, it is possible. Otherwise, questionable.*
- **There were two ways the surface water samples were collected. The CTD, when the ship was kept stable for the stations where the depth sampling was being done and from a bucket thermometer when the ship was moving.**
- **The salinity values were measured from the CTD during the stations and 6 hourly using an AutoSalinometer during the whole period of the expeditions taking samples from the bucket thermometer.**
- **For the depth sampling stations, the salinity and the isotopic composition is from the same sample. Whereas, in most of the cases, when the ship was moving the surface water samples for mesuring the isotopic composition of the are not same as the ones for which the salinity was measured. Nevertheless, the surface salinity values from the Autosalinometer which were measured along the sampling transect have been plotted in Fig. 4a.**
- **The salinity values have been added and the sample with a negative was in fact collected near the iceberg. The other probable reason for the negative value is the freshwater mixing of precipitation as this sample was collected during the passage of a low pressure system.**

*Detailed (mostly minor or editorial) comments:*

*Line 53: '. . . shows the sampling locations. ' How is Swv collected: at which height above the sea surface? How is spray avoided whether it is for Swv or Nwv. . .) .The detail of the collection method could be key for the results obtained. Later, it is mentioned that Swv samples could only be collected by fair weather. What kind of wind/swells/sea states are conductive to this measurement. All these relevant informations should be provided in the supplementary document.*

- **Swv samples have not been discussed in the modified version due to the reasons stated before.**
- **This information has been included in the supplementary document.**

*l. 72 '. . ., the change between trajectories corresponding to trade winds from the ones corresponding to westerlies happened at 31 ◦ S, whereas at 630S, a change. . .'*

- **Done**

*l. 106: the sentence 'The role of sst in governing d-excess. . .' This is only direct role, there is the indirect one of the air water mass and dependency on temperature.*

- **Noted. The sentence has been rephrased**

l. 113: 'The strength of the correlation is slightly higher. . .'

- **Done**

*l. 120: more sensitivity of ïAd'H to wind speed is not what comes out clearly from figure 5b. It might be a scaling issue of the ïAd'18O versus ïAd'D (note that R-square are small for both variables)*

- **Noted, the sentence has been omitted.**

*Line 129, the slope is the opposite in this latitude range between the two expeditions. This illustrates, I believe, either very different weather conditions (as seen in r of figure 2) or some issue somewhere, which would need to be further clarified*

- **This was a typing error the values of slopes are in fact same for both the expeditions.**

*Line 132? Sentence?*

- **The sentence has been rephrased**

*l. 134: '. . . complement. . .'*

- **Done**

*Line 135: correlation with SST of d18OSwv. Interesting, but these samples only taken when sea state not strong. Relating the dxs of seawater with relative humidity is a bit strange, as this is close to sea surface but with little wind/sea. I am not sure of what that brought?*

- **As explained earlier, the conditions deemed feasible for sampling close to the surface were constrained by the ocean surface conditions. In order to completely minimise the influence of sea spray on the water vapour samples. As mentioned before, since the meteorological measurements were done at the height of 15m the discussion on Swv samples hasn't been included in the modified version of the manuscript.**
- **We haven't compared the dxs of seawater with relative humidity.**

*l. 140: '. . . to the near-surface water vapour. . .' I am wondering whether the regression is significant, and in that case, whether this sentence is not anecdotical, and should be omitted.*

- **The sentence has been omitted.**

*In description of UCG (l. 157-191), the ratio term is ï A ¸ğ-h, which makes sense in this context. However, note that at saturation (fog, for example), thre is a definition issue. For that, The near saturation values reported on Figure 2 almost everywhere south of 50 ◦ S during SOE-IX is a major issue. How was this dealt with (and again, what confidence does one have in these near-saturated conditions, not witnessed the second year).*

- **As mentioned before, these were the conditions observed during the passage of low pressure systems as seen from the atmospheric pressure measurements.**

*Line 175: remove a 'same'. After that, the global closure is assumed as in Merlivat (1978)*

- **Done**

l. 201. I would mention the caveat of the issue of sea spray for high winds. Evaporation from sea spray is a large contribution to total evaporation in these conditions, and follows different laws (in the extreme case of all sea spray evaporating, this would for example yield Rev=Rl) There is also the caveat of below freezing temperature, and low SST close to sea ice formation temperature (just below fresh water freezing points), but I gather from figure 2 that this almost did not happen (correct?)

- **We have no way of knowing the fraction of sea spray contributing to the local moisture or whether all the sea spray that is formed is evaporating. So we have not discussed the influence of sea spray.**

*l. 219-220. This sentence is key. I think that this should be presented earlier.*

- **Done**

*l. 229: '. . . where westerlies dominate. . .'*

- **Done**

*L. 233: end of sentence missing*

- **Sentence rephrased**

*l. 236 '. . . is less, and is largely local. . .'*

- **Done**

---

## Author Comment (AC2) · 28 May 2020

We thank the reviewers for their valuable comments and suggestions. We have incorporated majority of their suggestions, which greatly improved the quality of the manuscript. The specific response to the reviewer's queries are addressed separately.

Below we listed the major changes that have been made in the revised version of the manuscript.
- The data referred to as Swv (19 samples) are not included in the revised manuscript. The meteorological parameters which were measured at the height of 15m above the surface of the ocean and not at the height at which Swv samples were collected (which was close to the ocean surface during calm conditions).
- The stucture of the paper is modified. The explanation of the the Unified and the Traditional Craig-Gordon models and the HYSPLIT back trajectory analysis has been moved to the methods section.
- The models are run for the turbulent indices (x) of 0-1 with an increment of 0.1. In the old manuscript we had run the models for only 0, 0.5 and 1.
- Most of the figures are modified to include the new results from modelling experiments and a few additional figures were included. All these figures with modified captions are presented below
- 4 new tables were also included in the revised submission. These tables along with the captions are presented below

[Figure]

Figure 1. The water vapor sampling locations shown as open circles overlain on the map of mean monthly sea surface temperature during the two expeditions. The sea surface temperature data is from Reanalysis dataset Kanamitsu et al. (2002) .

[Figure]

Figure 2. Latitudinal variability of measured meteorological parameters, temperature, relative humidity, wind speed and atmospheric pressure along the sampling transect. Filled blue diamonds and open circles in the temperature plot represent the sea surface temperature and air temperature respectively.

[Figure]

Figure 4. a) Measured $\delta^{18}O_{sw}$ as black filled circled and values of surface water isotopic composition extracted from the global sea water $\delta^{18}O$ database along the latitudinal transect (open black circles). Also plotted as orange filled squares are the salinity values along the transect b) Pink and purple filled diamonds depict the $\delta^{18}O$ of water vapor samples collected during the SOE-IX and SOE-X respectively at height of ≈ 15m above the water surface. c) latitudinal variation of dxs in water vapor samples shown as open red and purple diamonds for SOE-IX and SOE-X respectively, d) variation of $\delta^{18}O$, $\delta^2H$ and d-excess of water vapor along the transect as box plots grouped by latitudes, based on the HYSPLIT back trajectories.

[Figure]

Figure 5. Linear regression for isotopic composition of water vapor and physical parameters(sea surface temperature, relative humidity and wind speed). Hollow red and blue squares represent the $\delta^{18}O$ and $\delta^{2}H$ respectively and the shaded areas depict the 95% confidence bands. The linear regression lines are shown as blue and red for $\delta^{2}H$ and $\delta^{18}O$ respectively. The slope and intercept of the linear regression equations along with data from Uemura et al. (2008) are listed in Table 2 for $\delta^{18}O$ and Table 3 for $\delta^{2}H$.

[Figure]

Figure 6. Regression plots for d-excess (hollow black squares) in water vapor and the meteorological conditions (relative humidity, sea surface temperature and wind speed). The shaded region depicts the 95% confidence bands of d-excess. The slope and intercept of the regression equations along with data from Uemura et al. (2008) are listed in Table 4.

[Figure]

Figure 7. Comparison between the latitudinal distribution of the measured water vapor $\delta^{18}O$ (black lines) and that predicted by the TCG and UCG models for different molecular diffusivity ratios and turbulence indices, shown as colored lines.

[Figure]

Figure 8. Comparison between the latitudinal distribution of the measured d-excess in water vapor (black lines) and that predicted by the TCG and UCG models for different molecular diffusivity ratios and turbulence indices shown as colored lines

[Figure]

Figure 9. Slope of the relative humidity vs d-excess for the UCG (a-c) and the TCG (d-f) model runs (filled black squares) and the observed slope value (grey band).

[Figure]

Figure 10. Latitudinal variation of the observed $\delta^{18}O$ (a) as filled black diamonds and d-excess (b) as filled black circles and the modelled values (colored open diamonds and circles) for the model runs where the observed slope is comparable to the modelled slope. The statistical analysis of the observed and the modelled regression is listed in Table 5.

[Figure]

Figure 11. Linear regression equations between relative humidity (A), sea surface temperature (B) and wind speed (C) and the d-excess of the best-fit model runs. The dark and light pink shaded regions depict the 95% confidence bands and 95% prediction bands respectively.

[revised manuscript text omitted]

---

## Author Comment (AC3) · 28 May 2020

*The submitted paper by Dar et al presents an analysis of isotopic data from southern latitudes. These data are compared with a set of models and associated coefficients. While worthwhile, a revision of the structure and error analysis of conducted here would greatly improve this submission.*

**The structure of the paper has been modified according to the suggestions and error analysis included. The standard errors in both slope and intercept have been mentioned where ever the regression analysis has been done. The root mean square errors and the standard error for the model vs observed comparisions have been included.**

*First, the overall structure of this paper is a bit jumbled. Much of the discussion material about the CG models should be moved into the methods section. Similarly, the discussion of how the HYSPLIT back trajectories were ran should also be described in the methods section.*

**The stucture has been modified according to the suggestions.**

*Second, the error analysis needs to be more fully documented. The testing of different model formulations and parameters is helpful, however the authors do not fully evacuate the errors and biases associated with each model. A more rigorous description of errors across all variables is needed. Finally, is there an optimal set of an parameters that others should use (i.e. what value of x and fractionation factors minimizes errors and bias)?*

- **The error and biases associated with each model are well known and described in the number of previous studies, for e.g. The isotopic composition of atmospheric vapour is a factor responsible for the larges uncertainty in the model. Since our observations are mostly over the ocean AT 15 M LEVEL we have taken the global closure assumption i.e. the isotopic composition of atmospheric vapour is equal to the isotopic composition of evaporating water.**
- **In the modified version the performance of the models evaluated for different turbulence indices (i.e. the ratios of molecular and turbulent diffusion). The models were run for the values of x ranging from 0-1 with an increment of 0.1. We found that while the MJ and CD diffusivity ratios perform equally well for both the models, the PW values show the greater difference between the observed and modelled values.**
- **We found that UCG MJ for x=0.8, UCG CD for x=0.6 and TCG MJ for x=0.6 and TCG CD for x=0.7 perform equally well within the uncertainty limits.**

*L4: Add latitudes numbers here.*
- **Done**

*L5: Reword the sentence that starts "The inter annual", its not clear what your trying to say*
- **The sentence has been rephrased**

L19: Nearly and your double tilde are redundant.
- **This has been removed.**

L25: Missing an 'A' at the beginning?
- **Done**

L52: Define what you mean here by boundary later? Where these really at the boundary layer?
- **The sentence has been removed as the Swv samples have been omitted from the modified version of the manuscript.**

L65: This text on HYSPLIT methods should move to section 2.
- **Done**

*L76: Careful with your terminology here. A positive delta value signifies that it was more enriched in heavy isotopes relative to VSMOW only.*
- **The sentence has been rephrased.**

*L110-L120 What are the significance and or uncertaintie3s of these regressions.*
- **This information has been added to the table**

*L132: Why not put the regression coefficients and stats from Figs 5 & 6 in a table*
- **Done**

*L143-201: This needs to all move to methods.*
- **Done**

*L197: Please directly state the numbers you used here for the diffusivities*
- **Done**

*L209: While differences between the slopes and intercepts are interesting, a more error though analysis should be conducted. What is the overall bias associated with each model, what are the root mean squared errors, etc.*
- **The standard errors and root mean squared errors have been added in form of a table (Table 5)**

---

## Author Response (AR2)

We thank the reviewers and the editior for taking time to go through comment on the manuscript. Their comments and suggestions have greatly improved the quality of the manuscript

We have prepared a response to the reviewers comments and suggestions.

The reviewer comments/suggestions are in Italics and the responses have been provided in bold.

*The authors have appropriately responded in their letter to most of my main concerns and have taken adequate steps in the paper to respond to them.*

*However, there are a few points that need further action so that the readers will be better able to get full benefit of this interesting study.*

*One important point is that what is compared in the 'observation-model' comparison is not obvious as written. I understood that it was the result of Craig-Gordon (or its extended version) with the 'global closure assumption' (at least, I thought that these were equations 7 and 18 which are used, and not 1 and 11). This should be specified, for example in the introduction on line 47 (and maybe in the captions for some of the corresponding figures).*

- **The equations used are eq 7 for the Traditional Craig Gordon model and eq 18 for the Unified Craig Gordon Model. This has been mentioned in the introduction of the modified version and the caption of Fig 7 and Fig 8.**

*I did not fully understand also what is done with Pfahl and Wernli's study. It is cited as if they were providing diffusivity coefficients. However, it is not what the 2009 paper is about. The values cited on line 136 are close to be the inverse of their non-equilibrium fractionation factors (a power-law m independent of wind speed, as discussed inPfahl and Wernli (2003) paper). The issue discussed in the paper is the dependence of the fractionation factor as a function of wind which is specific to each study (Merlivat and Jouzel (1979), Cappa et al (2003) and Pfahl and Wernli (2003). In this sense, discussing using Pfahl and Wernli is different from just a question of diffusivity coefficients, for which I am only aware of the two studies (Merlivat, 1978; Cappa et al., 2003). It is in itself a parameterization. How is this taken into account here?*

- **In Pfahl and Wernli 2009, A comparison of the simulated deuterium excess with the measurements , the numerical values of the diffusivity ratios were calculated that lead to the best agreement between the observed and the simulated values. The best agreement was based on the r2 value for observed vs simulated d-excess in their study which they suggested to be used to calculate the isotopic composition of evaporation. The Merlivat, 1978; Cappa et al., 2003 diffusivity values were based on experimental studies while in the Pfahl and Wernli 2009 these values were calculated based on the r2 value of the simulated vs observed d-excess.**

*what is variable 'x' is not really defined (cf lines 123 and 136: the definition of 'x'). As there is no equation for x, it is hard to estimate quantitatively what is the index x. A definition equation should be introduced once, also better defining the index and how it is used.*

- **x is the turbulence index of atmosphere which signifies the proportion of vapor that escapes by isotopic fractionating molecular diffusion and non-fractionating**

> turbulent diffusion. When x = 1 the vapor escapes solely by molecular diffusion and for x = 0 the vapor escapes only due to turbulent diffusion.

- **This has been added to the modified version**

*The conclusion is rather short, and should be slightly expanded. I still think that one can mention there possible caveats (or modifications) to TCG or UCG models used for the evaporative flux. For example, sea spray formation and evaporation by very high winds are not taken into consideration. Same for condensation/deposition close to sea-ice at near-freezing temperatures (but probably only encountered at the most southern latitudes during these cruises). The advective mixing model is a very nice addition, nonetheless there can be other sources of advective humidity to the surface layers than from Antarctica (such as from upper atmospheric layers with different properties, in particular because of condensation/precipitation...) or reevaporation of rainfall. I am not saying by that these are necessarily important to take into consideration (after all, there still exists an average misfit wit hteh evaporation model both ind18O and d-excess on Figure 9, even north of 60°S). Also, the non-local source of the evaporation was discussed (with the back trajectories). This could be commented upon in the conclusion section (difference between local and 'non-local' sources.*

- **The conclusion is expanded in the revised version.**

*Also, there were some statistical relations which hold better when separating the two cruises. Any idea why?*

- **This is probably due to the difference in the number of points used to calculate the regression parameters and the increased scatter when the both the cruises are considered together. Nonetheless the differences are not too large.**

*Detailed comments:*
*In the reply letter, it is mentioned that some samples are from bucket collection, other from CTD cast. For buckets, could there be possible biases (in S and d18O, both too high), but also the two samples (S and d18O) are not collected with the same bucket. This should mentioned in the S table caption (adding for example a * to the salinities not collected from same bucket as the sample for d18O analysis). Otherwise, there are some anomalies in d18O/S that cannot be understood. For example; the low value d18O near an iceberg is not associated with low- S , which I would have assumed for this season, and if there no refreezing along the iceberg at depth (in this case, is it the S-value from a CTD cast or from a bucket; and is there or not refreezing, which would be interesting per se).*

- **The salinity values included in the modified version are for the samples only from the bucket thermometer . We have not included the CTD salinity values. The bucket sampling was done every 6 hours during the expeditions along the transect which sometimes didn't coincide with the water vapor sampling, therefore a different bucket sample was used to collect the sample for 18O measurement for which we didn't measure the salinity. These samples have been marked in the revised version.**

*At the end of the introduction, line 47, '... and different fractions of molecular vs turbulent diffusion.' I think that it is important to add 'in the framework of the global closure*

*assumption' (at least, I understood that these were equations 7 and 18 which are used, and not 1 and 11; this needs to be specified)*

- **As mentioned earlier, the equations used are eq.7 and eq 18. This has been specified and suggestion included in the modified version.**

*Line 96, suggestion to replace starting at 'ABove the ocean one can assume...' by 'The global closer assumption is commonly done, by which ...'*

- **This sentence has been rephrased according to the suggestion.**

*Line 136: I would change citation to Pfahl and Wernli.*

- **Done**

*Fig. 1: add year and dates in caption of fig. 1*

- **The caption has been modified.**

*Fig. 4: the end of the caption is unclear, as well as the response in the response letter. There may be a need to specify more how the data are grouped based on the trajectories (is the source latitude three days before considered, for example?)*

- **The plot (d) has been removed from the modified Fig 4.**

*FIgure 10: interesting, but I find what is written along the horizontal axis hard to read. Also, always the same sign (except for slope middle pattern). Nothing cut? Any idea why?*

- **The text of the horizontal axis has been enlarged and it is clear now.**
- **This plot depicts the differences between the observed and modelled lope and intercept of the meteorological parameters and d-excess. While the difference between the modelled and the observed slopes is the less, the models is all the relationships underpredict the intercept values. This due to the combination of molecular diffusivity and turbulence index values used in the Craig-Gordon equations.**

*Figure 11: why the choice of a blue column, and not just blue dots? This would be more consistent with earlier figures and also easier to visualize.*

- **The figure has been modified according to the suggestion.**

*l. 151: remove 'were'*

- **Done**

*l. 152: replace 'like' by 'such as'*

- **Done**

*l. 167: remove 'caused'*

- **Done**

*l. 170, and L. 187: SST instead of sst*

- **Done**

*l. 197: 'that' instead of 'than'*

- **Done**

*l. 215: 'd-excess of' to be replaced by 'and d-excess'*

- **Done**

*l. 217: 'observations' and later 'links'*

- **Done**

*l. 225: 'that predict'*

- **Done**

*l. 229: end of sentence 'are considered'?*
- **Done**

*l. 234: '... is insufficient...'*
- **Done**

*l. 235: remove 'The process like' and start sentence by 'Advective mixcing...'*
- **Done**

*l. 253: replace 'the contribution of which' by 'its contribution....' or something equivalent*
- **Rephrased**

*Suppl. Material:*
*for humidity, sling psychrometer used. What is the accuracy expected for its reading?*
*SST from bucket thermometer. The authors mention that it is accurate to 0.2°C? How is the bucket collected and its temperature read when there is high wind (more risk of cooling... evaporation?) I dont think that it will be as acurate with winds of 25 m/s or more that were sometimes encountered.*
- **There was a mistake in the accuracy of the psychrometer and the bucket thermometer. The values were interchaged. The accuracy of the bucket thermometer is $0.5^0$ while for the sling psychrometer $0.2^0$C. This has been corrected in the modified supplementary file.**
- **Empirical Relation used to calculate the Relative Humidity with expected accuracy of 2- 3% with accuracy 0.2° C of the sling Psyhcrometer.**
- **The bucket thermometer is lowered with the rope attached to it until it is immersed in the water for 5-10 minutes to get well mixed water. The bulb of the thermometer is at the bottom of the bucket and hence evaporative cooling due to high wind speed will happen at the surface and will not influence the measurements.**

*In table 4, the dates starting in the middle of the table invert month and day. There is also an incorrect date one line before the end.*
- **These have been corrected in the modified version.**

[revised manuscript text omitted]

Surface water samples were collected form Conductivity Temperature Depth (CTD) rosette when it was deployed and from a bucket thermometer used for measuring the sea surface temperature. Surface water samples were collected in 50ml High-Density Polyethylene air tight bottles.

All these samples were shipped to Bangalore for isotopic analysis and the measurements were carried out at the Centre for Earth Sciences, Indian Institute of Science, Bangalore. The protocol followed for the analysis of the gases after equilibration using a Finnigan Gas-bench II attached to a MAT 253 mass spectrometer is described in the (Rangarajan and Ghosh, 2011). For oxygen isotope analysis $200\mu L$ of water was transferred into an exetainer vial capped with butyl rubber septa and equilibrated with gas mixture 3% CO2+97% He for a period of 20 hours. For hydrogen isotopes, the water sample was equilibrated with gas mixture of 3% H2+97% He in presence of platinum catalyst (Hokko bead sticks) for a period of 80mins. The isotope ratios are expressed in ‰ using the standard $\delta$ notation relative to Vienna Standard Mean Ocean Water (VSMOW). Internal laboratory standards (OASIS-WWW, OASIS-LDK and OASIS-VOULEP) calibrated against the international water standards (VSMOW, Standard Light Antarctic Precipitation, and Greenland Ice Sheet Project) available from International Atomic Energy Agency in Vienna, were used to determine the accuracy and precision of the analysis. To account for intra batch calibration and drift correction, additional internal laboratory standards were measured in a batch. The overall analytical uncertainty on the measurements ($\pm 1\sigma$), as determined from replicate measurements of internal laboratory standards, were respectively $\pm 1.0$‰ and $\pm 0.1$‰ for $\delta^2 H$ and $\delta^{18}O$. Isotopic values are reported here with one standard deviation.

**2  Meteorological measurements**

Atmosphere readings were taken via multiple instruments on-board the ocean research vessel SA Agulhas. Relative Humidity was calculated from the Psychrometric charts with the help of dry bulb and wet bulb temperature readings from sling Psychrometer with a range of -5°C to +50°C and a least count of 0.2°C. The expected accuracy in the relative humidity measurements from the psychrometer in 2-3%. Air temperature, Atmospheric Pressure, Wind's magnitude and Direction, GPS were logged from AWS (Automatic Weather Station) installed on board the ship. Salinity was measured using an Auto Salino Meter (Tsurumi Seiki Co. Ltd, Japan. Salinity values are expressed in the 1978 Practical Salinity Scale (PSU) (PSS-78) with a precision of $\pm 0.005$ PSU. Sea Surface Temperature (SST) was measured using a bucket thermometer (Theodor Friedrichs and Co, Germany; accuracy $\pm$ 0.5°C).

**3 Supplementary figures and tables**

[Figure]

**Figure 1. (S)** Slope of the relative humidity vs d-excess for the UCG (a-c) and the TCG (d-f) model runs (filled black squares) and the observed value (grey band).

[Figure]

**Figure 2. (S)** Linear regression equations between relative humidity (A), sea surface temperature (B) and wind speed (C) and the d-excess of the best-fit model runs. The dark and light pink shaded regions depict the 95% confidence bands and 95% prediction bands respectively.

**Table 1. (S)** Slope, intercept and $r^2$ of the linear regression equations between meteorological parameters (relative humidity, sea surface temperature and winds speed) and $\delta^{18}O$ for different sample classifications. Also listed are the regression parameters for the data from Uemura et al. (2008)

| Met. vs $\delta^{18}O$ | Classification | Intercept | | Slope | | Statistics |
|---|---|---|---|---|---|---|
| | | Value | Standard Error | Value | Standard Error | R-Square(COD) |
| Relative Humidity | ALL | -11.43 | 3.43 | -0.06 | 0.04 | 0.03 |
| | ALL North of 65°S | -15.49 | 2.04 | 0.02 | 0.03 | 0.01 |
| | ALL South of 65°S | -11.05 | 5.11 | -0.12 | 0.06 | 0.15 |
| | SOE IX North of 65°S | -12.37 | 2.59 | -0.02 | 0.03 | 0.01 |
| | SOE X North of 65°S | -18.95 | 3.39 | 0.06 | 0.04 | 0.07 |
| | Uemura All | -20.61 | 2.81 | 0.05 | 0.04 | 0.02 |
| Sea Surface Temperature | ALL | -18.43 | 0.53 | 0.27 | 0.05 | 0.33 |
| | ALL North of 65°S | -15.47 | 0.40 | 0.12 | 0.03 | 0.27 |
| | ALL South of 65°S | -19.38 | 0.71 | -2.37 | 0.62 | 0.41 |
| | SOE IX North of 65°S | -15.30 | 0.70 | 0.12 | 0.04 | 0.26 |
| | SOE X North of 65°S | -15.52 | 0.51 | 0.11 | 0.05 | 0.19 |
| | ALL SOE X | -16.82 | 0.42 | 0.19 | 0.05 | 0.33 |
| | ALL SOE IX | -21.07 | 0.96 | 0.41 | 0.07 | 0.51 |
| | Uemura All | -17.40 | 0.46 | 0.19 | 0.05 | 0.16 |
| Wind Speed | ALL | -17.85 | 1.01 | 0.10 | 0.06 | 0.04 |
| | ALL North of 65°S | -12.74 | 0.60 | -0.09 | 0.03 | 0.13 |
| | ALL South of 65°S | -23.76 | 1.40 | 0.25 | 0.11 | 0.21 |
| | SOE IX North of 65°S | -12.62 | 0.77 | -0.07 | 0.05 | 0.11 |
| | SOE X North of 65°S | -13.05 | 0.98 | -0.09 | 0.05 | 0.12 |
| | ALL SOE X | -16.66 | 0.93 | 0.06 | 0.05 | 0.03 |
| | ALL SOE IX | -18.77 | 1.80 | 0.14 | 0.12 | 0.04 |

**Table 2. (S)** Slope, intercept and $r^2$ of the linear regression equations between meteorological parameters (relative humidity, sea surface temperature and winds speed) and $\delta^2 H$ for different sample classifications. Also listed are the regression parameters for the data from Uemura et al. (2008)

| Met. vs $\delta^2 H$ | Classification | Intercept | | Slope | | Statistics |
|---|---|---|---|---|---|---|
| | | Value | Standard Error | Value | Standard Error | R-Square(COD) |
| Relative Humidity | ALL | -57.14 | 27.50 | -0.90 | 0.35 | 0.09 |
| | ALL North of 65°S | -77.52 | 18.79 | -0.42 | 0.24 | 0.06 |
| | ALL South of 65°S | -41.57 | 22.20 | -0.77 | 0.27 | 0.27 |
| | SOE IX North of 65°S | -102.94 | 28.84 | -0.18 | 0.38 | 0.01 |
| | SOE X North of 65°S | -133.31 | 30.61 | 0.10 | 0.41 | 0.00 |
| | Uemura All | -110.71 | 22.14 | -0.16 | 0.28 | 0.00 |
| Sea Surface Temperature | ALL | -149.02 | 3.84 | 2.76 | 0.34 | 0.49 |
| | ALL North of 65°S | -128.58 | 2.78 | 1.68 | 0.20 | 0.61 |
| | ALL South of 65°S | -127.93 | 5.09 | 1.76 | 0.31 | 0.60 |
| | SOE IX North of 65°S | -128.31 | 3.31 | 1.41 | 0.30 | 0.50 |
| | SOE X North of 65°S | -136.73 | 2.72 | 1.95 | 0.30 | 0.55 |
| | ALL SOE X | -168.91 | 7.05 | 3.88 | 0.53 | 0.63 |
| | ALL SOE IX | -154.43 | 5.43 | -17.47 | 4.77 | 0.39 |
| | Uemura All | -135.09 | 2.99 | 2.28 | 0.36 | 0.41 |
| Wind Speed | ALL | -133.39 | 8.50 | 0.36 | 0.51 | 0.01 |
| | ALL North of 65°S | -90.28 | 5.20 | -1.24 | 0.29 | 0.29 |
| | ALL South of 65°S | -85.84 | 6.89 | -1.19 | 0.41 | 0.29 |
| | SOE IX North of 65°S | -98.47 | 7.54 | -1.04 | 0.40 | 0.23 |
| | SOE X North of 65°S | -127.16 | 7.58 | 0.07 | 0.43 | 0.00 |
| | ALL SOE X | -138.60 | 15.43 | 0.63 | 1.01 | 0.01 |
| | ALL SOE IX | -182.36 | 11.09 | 1.49 | 0.85 | 0.13 |

**4 Data used in this study**

**Table 3. (S)** SOE-IX meteorological data, water vapor and surface water isotopic composition.

| Date | Lon | Lat | Tair ($^0C$) | Atm. Pres. ($mbar$) | Rel. Hum. (%) | Wind Speed ($m/s$) | SST ($^0C$) | $\delta^{18}O$ (‰) | $\delta^2H$ (‰) | d-excess (‰) | Sal. ($PSU$) | $\delta^{18}O_{SW}$ (‰) |
|---|---|---|---|---|---|---|---|---|---|---|---|---|
| 08/01/2017 | 57.50 | -27.38 | 29.60 | 1015.80 | 69.31 | 9.68 = | 27.0 | -11.97 | -83.04 | 12.71 | 35.59 | |
| 08/01/2018 | 57.52 | -28.66 | 29.20 | 1014.80 | 68.69 | 4.80 | 27.0 | -12.00 | -80.79 | 15.17 | 35.50 | |
| 09/01/2017 | 57.49 | -31.53 | 26.13 | 1016.50 | 69.31 | 7.58 | 25.0 | -11.92 | -82.11 | 13.22 | 35.54 | 0.31 |
| 09/01/2017 | 57.50 | -32.26 | 24.13 | 1016.50 | 77.01 | 14.25 | 24.0 | -12.16 | -84.65 | 12.66 | 35.57 | |
| 09/01/2017 | 57.51 | -33.44 | 19.83 | 1018.67 | 75.15 | 9.82 | 21.0 | -12.22 | -87.82 | 9.98 | 35.41 | |
| 10/01/2017 | 57.50 | -35.38 | 21.50 | 1019.67 | 73.54 | 10.75 | 19.5 | -12.29 | -95.66 | 2.64 | 35.47 | 0.33 |
| 10/01/2017 | 57.51 | -36.43 | 21.17 | 1017.67 | 82.94 | 5.20 | 19.0 | -10.86 | -82.74 | 4.16 | 35.46 | |
| 11/01/2017 | 57.87 | -39.11 | 13.88 | 1018.25 | 72.25 | 27.25 | 17.0 | -14.62 | -104.60 | 12.36 | 35.58 | |
| 12/01/2017 | 58.41 | -40.07 | 13.67 | 1027.83 | 60.86 | 9.00 | 16.0 | -15.57 | -107.66 | 16.89 | 35.41 | |
| 12/01/2017 | 57.94 | -40.08 | 12.38 | 1028.00 | 67.06 | 7.75 | 16.5 | -15.91 | -109.99 | 17.33 | 35.51 | |
| 14/01/2017 | 58.52 | -40.09 | 17.40 | 1017.80 | 82.36 | 25.40 | 16.0 | -11.96 | -91.18 | 4.52 | 35.36 | 0.38 |
| 13/01/2017 | 57.99 | -40.21 | 13.00 | 1025.00 | 59.18 | 11.00 | 16.5 | -14.88 | -100.35 | 18.65 | 35.37 | |
| 15/01/2017 | 59.46 | -41.38 | 14.17 | 1022.00 | 67.80 | 15.17 | 16.5 | -14.37 | -96.94 | 18.03 | 35.47 | 0.35 |
| 16/01/2017 | 61.15 | -43.67 | 16.53 | 1023.83 | 71.16 | 7.00 | 17.0 | -14.78 | -101.41 | 16.85 | 35.49 | -0.26* |
| 16/01/2017 | 62.72 | -45.46 | 11.67 | 1012.33 | 91.67 | 15.33 | 12.0 | -12.37 | -100.60 | -1.60 | 35.52 | |
| 17/01/2017 | 64.00 | -47.02 | 7.33 | 1003.33 | 93.33 | 20.00 | 7.0 | -15.19 | -129.06 | -7.57 | 33.72 | -0.12 |
| 18/01/2017 | 64.10 | -49.02 | 5.60 | 1008.60 | 88.86 | 20.80 | 5.5 | -13.97 | -118.99 | -7.20 | 33.52 | -0.12 |
| 19/01/2017 | 64.17 | -51.05 | 5.45 | 1011.67 | 78.70 | 21.17 | 5.0 | -14.40 | -117.37 | -2.18 | 33.80 | -0.07 |
| 19/01/2017 | 67.00 | -51.73 | 4.79 | 1009.29 | 98.97 | 8.57 | 5.0 | -12.97 | -106.26 | -2.52 | 33.69 | |
| 20/01/2017 | 68.49 | -54.01 | 4.54 | 1006.57 | 93.73 | 15.57 | 4.5 | -12.68 | -101.07 | 0.37 | 33.87 | 0.05* |
| 21/01/2017 | 69.29 | -57.40 | 4.00 | 996.25 | 96.39 | 7.25 | 3.0 | -13.39 | -101.24 | 5.90 | 33.89 | 0.03* |
| 22/01/2017 | 70.08 | -61.95 | 0.98 | 970.20 | 100.00 | 21.00 | 2.0 | -19.46 | -162.93 | -7.26 | 33.61 | -0.07 |
| 23/01/2017 | 68.34 | -64.00 | 0.14 | 976.17 | 98.31 | 39.67 | 0.5 | -14.82 | -126.91 | -8.37 | 33.83 | 0.18* |
| 24/01/2017 | 74.01 | -65.99 | 0.09 | 983.00 | 95.57 | 14.86 | 0.5 | -20.35 | -160.07 | 2.72 | 33.86 | |
| 25/01/2017 | 72.54 | -67.96 | -0.51 | 992.14 | 96.45 | 20.57 | 1.5 | -22.62 | -182.23 | -1.28 | 33.79 | -0.61 |
| 26/01/2017 | 74.01 | -67.99 | 1.00 | 991.33 | 72.19 | 7.67 | 1.5 | -22.79 | -177.95 | 4.41 | 33.43 | |
| 26/01/2017 | 74.00 | -68.00 | -0.33 | 992.00 | 83.67 | 12.00 | 1.5 | -19.94 | -154.85 | 4.66 | 33.21 | |
| 31/01/2017 | 76.00 | -68.00 | -1.75 | 978.00 | 94.23 | 5.83 | 1.5 | -24.86 | -193.54 | 5.31 | 32.73 | |
| 27/01/2017 | 74.05 | -68.02 | 1.08 | 990.00 | 65.92 | 5.67 | 1.5 | -23.33 | -182.05 | 4.56 | 33.38 | |
| 30/01/2017 | 76.12 | -68.04 | -0.50 | 988.50 | 84.14 | 14.25 | 1.0 | -22.17 | -174.22 | 3.11 | 32.77 | |
| 27/01/2017 | 73.93 | -68.21 | 0.12 | 988.40 | 80.13 | 8.20 | 2.0 | -27.06 | -221.38 | -4.87 | 32.42 | |
| 28/01/2017 | 74.01 | -68.60 | -2.10 | 987.00 | 90.36 | 5.40 | 2.0 | -27.47 | -216.45 | 3.28 | | |
| 31/01/2017 | 75.90 | -69.19 | -0.92 | 983.67 | 100.00 | 12.67 | 2.5 | -24.25 | -184.09 | 9.94 | 33.74 | |
| 01/02/2017 | 76.05 | -69.34 | 0.33 | 991.33 | 97.92 | 7.33 | 0.0 | -27.14 | -211.65 | 5.46 | 32.07 | |
| 01/09/2018 | 74.73 | -66.78 | 0.12 | 978.42 | 78.87 | 25.04 | 0.0 | -17.54 | -131.63 | 8.72 | 32.73 | -0.57 |
| 01/02/2018 | 73.31 | -66.80 | 0.74 | 989.95 | 76.64 | 8.75 | 0.0 | -18.48 | -142.40 | 5.41 | 33.37 | -0.69 |

\* The isotopic composition and salinity values of the surface ocean water are from different bucket samples.

**Table 4. (S)** SOE-X meteorological data, water vapor and surface water isotopic composition.

| Date | Lon | Lat | Tair ($^0C$) | Atm. Pres. (mbar) | Rel. Hum. (%) | Wind Speed (m/s) | SST ($^0C$) | $\delta^{18}O$ (‰) | $\delta^2H$ (‰) | d-excess (‰) | Sal. (PSU) | $\delta^{18}O_{SW}$ (‰) |
|---|---|---|---|---|---|---|---|---|---|---|---|---|
| 12/10/2017 | 57.56 | -21.98 | 26.13 | 1015.50 | 75.00 | 11.50 |  | -12.34 | -94.50 | 4.20 |  |  |
| 12/11/2017 | 57.79 | -26.80 | 28.55 | 1012.70 | 71.60 | 2.70 | 25.0 | -11.46 | -88.03 | 3.61 | 35.41 | 0.53 |
| 12/12/2017 | 58.00 | -31.05 | 21.53 | 1016.00 | 74.53 | 16.76 | 21.5 | -12.87 | -101.43 | 1.52 | 35.42 | 0.77 |
| 13/12/17 | 58.20 | -35.24 | 19.00 | 1015.00 | 55.75 | 10.93 | 21.0 | -15.58 | -110.07 | 14.54 | 35.56 | 0.36 |
| 14/12/17 | 58.49 | -39.84 | 18.00 | 1007.13 | 80.82 | 7.72 | 16.5 | -12.21 | -98.43 | -0.77 | 35.51 | 0.36 |
| 15/12/17 | 57.49 | -39.99 | 14.46 | 995.03 | 84.68 | 17.36 | 16.5 | -13.54 | -100.92 | 7.39 | 33.84 | 0.61 |
| 16/12/17 | 58.80 | -40.18 | 14.05 | 1015.04 | 63.00 | 18.29 | 16.0 | -15.56 | -110.40 | 14.11 | 35.47 | 0.28 |
| 17/12/17 | 58.38 | -40.19 | 16.93 | 1016.80 | 77.68 | 23.94 | 16.5 | -12.11 | -97.43 | -0.54 | 35.46 |  |
| 18/12/17 | 60.50 | -42.89 | 9.33 | 1011.52 | 57.95 | 24.82 | 11.0 | -15.52 | -115.67 | 8.51 | 33.95 |  |
| 19/12/17 | 62.63 | -45.69 | 8.07 | 1015.17 | 55.04 | 9.29 | 8.0 | -16.56 | -118.22 | 14.29 | 34.38 | 0.08 |
| 20/12/17 | 64.35 | -48.07 | 6.94 | 991.67 | 80.92 | 23.65 | 5.0 | -13.62 | -112.24 | -3.28 | 33.90 | -0.41 |
| 21/12/17 | 63.85 | -50.78 | 4.52 | 972.07 | 78.00 | 27.25 | 4.5 | -14.31 | -120.64 | -6.12 | 33.81 | -0.55 |
| 22/12/17 | 65.58 | -53.07 | 4.79 | 970.60 | 70.21 | 7.64 | 4.0 | -14.58 | -121.16 | -4.54 |  | -0.02 |
| 23/12/17 | 68.23 | -54.02 | 2.26 | 981.60 | 82.06 | 29.24 | 3.0 | -15.81 | -132.28 | -5.76 | 33.90 | 0.02* |
| 24/12/17 | 69.03 | -56.43 | 2.80 | 993.74 | 78.40 | 13.49 | 2.5 | -14.05 | -118.57 | -6.19 | 33.95 | -0.01* |
| 25/12/17 | 70.14 | -58.03 | 3.18 | 1002.36 | 70.17 | 16.68 | 2.0 | -14.44 | -111.16 | 4.33 | 33.92 | -0.57 |
| 25-26/12/17 | 70.12 | -59.05 | 1.80 | 1002.17 | 82.14 | 15.67 | 0.5 | -13.58 | -108.05 | 0.60 | 33.08 | -0.39 |
| 26/12/17 | 71.59 | -59.99 | 1.21 | 993.15 | 83.82 | 13.50 | 0.0 | -13.35 | -112.80 | -5.97 | 33.61 | -0.48 |
| 26/12/17 | 71.14 | -61.06 | 0.14 | 984.87 | 88.64 | 19.45 | 0.5 | -13.93 | -120.74 | -9.32 | 33.68 | -0.51 |
| 27/12/17 | 70.90 | -61.66 | 1.19 | 985.72 | 79.35 | 6.12 | 0.5 | -15.74 | -129.25 | -3.30 | 33.71 | -0.20 |
| 17-18/01/18 | 57.49 | -61.99 | 2.80 | 986.17 | 74.25 | 20.14 | 1.5 | -15.39 | -130.38 | -7.28 | 33.70 |  |
| 28/12/17 | 69.99 | -63.01 | -1.02 | 990.34 | 74.22 | 22.52 | 1.5 | -21.18 | -163.28 | 6.18 | 33.63 | -0.59 |
| 17/01/18 | 57.52 | -63.05 | 1.16 | 974.27 | 81.60 | 29.23 | 1.0 | -16.63 | -141.06 | -8.01 | 33.49 |  |
| 16-17/01/18 | 57.42 | -64.01 | 1.90 | 969.20 | 83.90 | 27.63 | 1.0 | -15.02 | -130.43 | -10.24 | 33.20 | -0.22* |
| 14/01/18 | 66.99 | -65.49 | -0.70 | 971.30 | 79.66 | 17.03 | 0.0 | -15.40 | -121.20 | 2.00 |  | -0.35 |
| 16/01/18 | 57.85 | -65.51 | 1.34 | 967.24 | 84.00 | 27.02 | 1.0 | -15.70 | -141.37 | -15.73 | 33.57 | -1.05* |
| 30/12/17 | 74.91 | -65.51 | -0.36 | 978.48 | 76.16 | 6.21 | 0.0 | -18.88 | -148.91 | 2.15 | 33.39 | -0.62 |
| 10/01/2018 | 68.81 | -65.51 | 0.31 | 980.96 | 61.30 | 4.00 | -0.5 | -20.40 | -151.04 | 12.15 | 33.69 | -3.45* |
| 31/12/17 | 73.84 | -65.52 | -0.40 | 984.47 | 74.00 | 14.10 | 0.0 | -18.33 | -152.82 | -6.18 | 33.51 | -1.16* |
| 31/12/17-1/1/18 | 72.67 | -65.54 | -0.25 | 988.00 | 81.50 | 16.40 | -1.0 | -16.77 | -157.86 | -23.71 | 33.61 | -0.80 |
| 15/01/18 | 57.26 | -65.58 | -0.33 | 973.95 | 72.66 | 11.08 | 0.5 | -19.64 | -155.75 | 1.40 | 33.48 |  |
| 29-30/12/17 | 74.79 | -66.35 | -1.35 | 982.00 | 62.50 | 7.00 | -0.5 | -21.11 | -161.02 | 7.88 | 32.99 | -0.64 |
| 07/01/2018 | 74.98 | -66.43 | 1.86 | 981.77 | 63.53 | 4.88 | 0.0 | -18.17 | -137.79 | 7.59 | 32.73 |  |
| 01/01/2018 | 73.00 | -66.45 | 0.90 | 987.10 | 77.67 | 5.60 | -0.5 | -17.65 | -144.89 | -3.72 | 32.73 | -0.33* |
| 09/01/2018 | 74.73 | -66.78 | 0.12 | 978.42 | 78.87 | 25.04 | 0.0 | -17.54 | -131.63 | 8.72 | 32.73 | -0.57 |
| 02/01/2018 | 73.31 | -66.80 | 0.74 | 989.95 | 76.64 | 8.75 | 0.0 | -18.48 | -142.40 | 5.41 | 33.37 | -0.69 |

50     * The isotopic composition and salinity values of the surface ocean water are from different bucket samples.